# Genetic evolution of keratinocytes to cutaneous squamous cell carcinoma

Bishal Tandukar [1,2,9], Delahny Deivendran [1,2,9], Limin Chen[1,2], Aravind K. Bandari[1,2], Noel Cruz-Pacheco [1,2], Harsh Sharma[1,2], Meng Wang [1,2], Albert Xu[1,2], Daniel B. Chen[1,2,3], Christopher D. George [4], Annika L. Marty [1,2,5], Raymond J. Cho [1], Jeffrey B. Cheng[1], Drew Saylor[1], Pedram Gerami[6], Iwei Yeh [1,2,7], Sarah T. Arron [8], Boris C. Bastian [1,2,7] & A. Hunter Shain [1,2] ✉

Cutaneous squamous cell carcinomas (cSCCs) arise from keratinocytes in the skin, but the molecular changes driving this transformation remain unclear. To better understand this process, we perform multi-omic profiling of keratinocytes, actinic keratoses, and cSCCs. Single-cell mutational analyses reveal that most keratinocytes have remarkably low mutation burdens; however, keratinocytes with *TP53* or *NOTCH1* mutations exhibit substantially higher burdens. These findings suggest that keratinocytes can withstand high dosages of cumulative ultraviolet radiation, but certain pathogenic mutations break these adaptive mechanisms, inducing a mutator phenotype. Mutational profiling of cSCCs adjacent to actinic keratoses reveals *TERT* promoter and *CDKN2A* mutations emerge in actinic keratoses, whereas additional mutations that inactivate *ARID2* and activate the mitogen-activated protein kinase pathway delineate the transition to cSCC. Surprisingly, actinic keratoses are often not related to their neighboring cSCC. Spatial analyses reveal gene expression heterogeneity, including checkpoint molecule enrichment at invasive fronts, highlighting tumor and immune cell interactions.

Cutaneous squamous cell carcinoma is the second most common type of cancer[1] and is responsible for an estimated 2500-15,000 deaths per year in the United States[2–4]. These estimates vary widely because there are no cancer registries to officially track the mortality of cutaneous squamous cell carcinoma, but this range is on par with melanoma, gastric cancer, cervical cancer, liver cancer, and kidney cancer[5]. Compared to other cancer subtypes with similar death tolls, the evolution of cutaneous squamous cell carcinoma remains poorly understood, posing a major obstacle towards improvement of prevention strategies and development of new therapeutic modalities.

Fully evolved cutaneous squamous cell carcinomas have somatic alterations disrupting the p53 and Notch signaling pathways[6]. To a lesser extent, they also have alterations known to activate the MAPK/ PI3-Kinase pathways, upregulate telomerase, perturb the SWI/SNF chromatin remodeling complex, abrogate cell-cycle checkpoint control, and/or deactivate the Hippo signaling pathway[6]. The order in which these somatic alterations become selected during tumor evolution is not entirely known, but sequencing of normal skin and precursor lesions, such as actinic keratoses, provide some insights.

Cutaneous squamous cell carcinoma arises from keratinocytes of the epidermis. Clonal patches of keratinocytes with p53 or Notch mutations can be found in the epidermis, increasing in density with higher levels of cumulative sun exposure[7–15]. Actinic keratoses are low-risk precursor lesions of cutaneous squamous cell carcinoma and

---

[1]Department of Dermatology, University of California San Francisco, San Francisco, CA, USA. [2]Helen Diller Family Comprehensive Cancer Center, University of California San Francisco, San Francisco, CA, USA. [3]School of Medicine, Case Western Reserve University, Cleveland, OH, USA. [4]Department of Dermatology, Erasmus MC, Rotterdam, Netherlands. [5]Institute of Physiology, University of Zurich, Zurich, Switzerland. [6]Department of Dermatology, Feinberg School of Medicine, Northwestern University, Chicago, IL, USA. [7]Department of Pathology, University of California San Francisco, San Francisco, CA, USA. [8]Peninsula Dermatology, Burlingame, CA, USA. [9]These authors contributed equally: Bishal Tandukar, Delahny Deivendran. ✉e-mail: hunter.shain@ucsf.edu

probably arise from these patches. While several studies have sequenced actinic keratoses[16–18], their genetic drivers remain incompletely understood due to their complex clonal architecture, as we discuss in more detail below.

In this work, we profile the mutational and transcriptional landscapes of individual keratinocytes from clinically normal human skin to better understand the genetic evolution of cutaneous squamous cell carcinoma. We also perform DNA sequencing and spatial transcriptomics of actinic keratoses that are adjacent to cutaneous squamous cell carcinomas. We reveal the key events driving the transformation of cutaneous squamous cell carcinomas from epidermal keratinocytes through the pre-neoplastic and pre-malignant stages of progression.

## Results

### Mutational landscapes of individual keratinocytes from normal human skin

We began by profiling the mutational landscapes of epidermal keratinocytes and comparing them to epidermal melanocytes and dermal fibroblasts from the same biopsies. It remains difficult to detect somatic mutations in an individual cell with high specificity and sensitivity[19–22]. To achieve this goal, we adapted a workflow, previously designed to genotype melanocytes at single-cell resolution[23], to also work on keratinocytes and fibroblasts (see Fig. S1a for an overview). Briefly, we clonally expanded individual skin cells ex vivo, producing small colonies of daughter cells (typically 200 cells per colony), extracted their DNA and RNA, and further amplified the nucleic acids in vitro. The combination of clonal expansion of cells and in vitro amplification of DNA/RNA produced sufficient template material to sensitively detect mutations. To call somatic mutations at high specificity, we identified patterns in the sequencing data[23], which can distinguish bona fide mutations from artifacts introduced during amplification.

A limitation to clonal expansion is that it may introduce a bias in the cells that grow out and are mutationally profiled. Such a critique would apply to other studies that measure the mutational landscapes of individual cells via clonal expansion[24–27], or more broadly, could be applied to any study that has ever performed tissue culture. It is a tradeoff that we believe is justified by the high-quality mutational calls, at single-cell resolution, that can be achieved with the help of clonal expansion. To minimize biases, we optimized culture conditions so that a high proportion of cells clonally expanded from each tissue (see methods). For example, we sorted cells with limited dilution in lieu of fluorescence-activated cell sorting (FACS), despite considerable effort required. Nevertheless, bias may persist, so we compared the mutational landscapes of individual cells to bulk sequencing measurements and discuss, below, how each approach may skew mutational calls.

In total, we measured somatic mutations in single-cell expansions of 137 keratinocytes, 131 melanocytes, and 23 fibroblasts from 22 different skin biopsies from 15 unique donors (Supplementary Data S1, Fig. S1b). The donors ranged in age from 35 to 95 years and included 14 individuals of European ancestry and 1 individual of admixed American ancestry (Fig. S2, Supplementary Data S2). Given the limited ancestral diversity in our cohort, it was not feasible to perform mutation comparisons between ancestral populations in the present study.

Skin was collected from body sites that experience different degrees of habitual sun exposure, including the buttocks, trunk, and head/neck area (Supplementary Data S1). The lineage of each cell was confirmed by its cytological features and gene expression profiles (Fig. S1c, d). Most clonal expansions were sequenced at exome resolution (95X coverage on average), including all keratinocytes.

The median mutation burden of keratinocytes was 1.14 mutations per megabase (mut/Mb), which was lower than the mutation burdens of melanocytes (3.91 mut/Mb) and fibroblasts (1.92 mut/Mb, Fig. 1a). These differences held up within most skin biopsies where multiple

cell types were sequenced, thus reflecting cell type variation rather than donor-to-donor variation. Keratinocytes from sun-exposed skin had higher mutation burdens than those from sun-shielded skin, but the differences were smaller than in other cell populations (Fig. S3a). For instance, keratinocytes from the upper back had a median mutation burden of 1.70 mut/Mb versus 0.38 mut/Mb for keratinocytes from the buttocks. By contrast, melanocytes from the upper back had a median mutation burden of 14.81 mut/Mb versus 0.30 mut/Mb for melanocytes from the buttocks (Fig. S3a). Curiously, cells from the upper back, irrespective of type, had higher mutation burdens on average than from the head/neck area, seemingly at odds with the cumulative doses of UV exposure typically experienced at these sites. However, this finding is consistent with previous observations by our group[23] and others[11], warranting future studies to understand how cells from chronically sun-exposed body sites, such as the head/neck, keep their mutation burdens relatively low.

The low mutation burden of keratinocytes compared to melanocytes and fibroblasts is unexpected. Keratinocyte stem cells and melanocytes both reside in the basal layer of the epidermis and are expected to receive similar doses of UV radiation. Fibroblasts reside in the underlying dermis and thus would be expected to receive lower doses of UV radiation than either keratinocytes or melanocytes.

While most keratinocytes had low mutation burdens, some had mutation burdens as high as 49.71 mutations/Mb (Fig. 1b, c). We annotated cells with mutations known to be pathogenic in cutaneous squamous cell carcinoma[6], and every keratinocyte with more than 3.5 mutations/Mb had at least one pathogenic mutation. Among these, keratinocytes with missense mutations in *TP53*, which are known to confer a dominant negative effect on the protein[28], had the highest mutation burdens. These findings suggest that wild-type keratinocytes (i.e., cells without pathogenic mutations) are remarkably well-adapted to maintain their mutation burdens in check.

It remains unclear how keratinocytes without pathogenic mutations maintain low mutation burdens in human skin. Previous studies[29–31], as well as our own in vitro UV irradiation experiments (Fig. S4a), have shown that melanocytes are less prone to undergo apoptosis in response to UV-induced DNA damage than keratinocytes. This differential sensitivity could, over time, result in the selective elimination of keratinocytes with higher mutation burdens, thereby enriching the population for cells with fewer mutations. However, under our experimental conditions, the surviving keratinocytes accumulated more mutations than the surviving melanocytes (Fig. S4b). It is important to note that our in vitro system involves two-dimensional cultures and models an acute UV dose, which may not accurately recapitulate the cumulative and spatially heterogeneous exposures experienced in intact skin. Future studies will be required to elucidate how keratinocytes maintain low mutation burdens over decades of environmental exposure.

Mutational signature analyses[32] revealed differences in the types of mutations between keratinocytes, melanocytes, and fibroblasts (Fig. 1b, S3b). SBS7a, which has been attributed to UV radiation, was present in keratinocytes; however, it contributed to a lower proportion of mutations than in melanocytes and fibroblasts (Fig. 1d). Keratinocytes, instead, had higher proportions of mutations with clock-like signatures- SBS1 and SBS5 (Fig. 1e), which are associated with aging and cumulative mitoses[33].

To better understand biases that may be introduced by our single-cell genotyping workflow, we compared the mutational landscapes of individual keratinocytes to bulk cell sequencing of epidermis. Encouragingly, the trinucleotide contexts of mutations from individual keratinocytes, aggregated together, were nearly identical to those observed in an independent study that performed bulk-cell sequencing of epidermis[11] (Fig. S5a).

While the types of mutations in our study were similar to bulk-cell measurements, the number of mutations per cell was

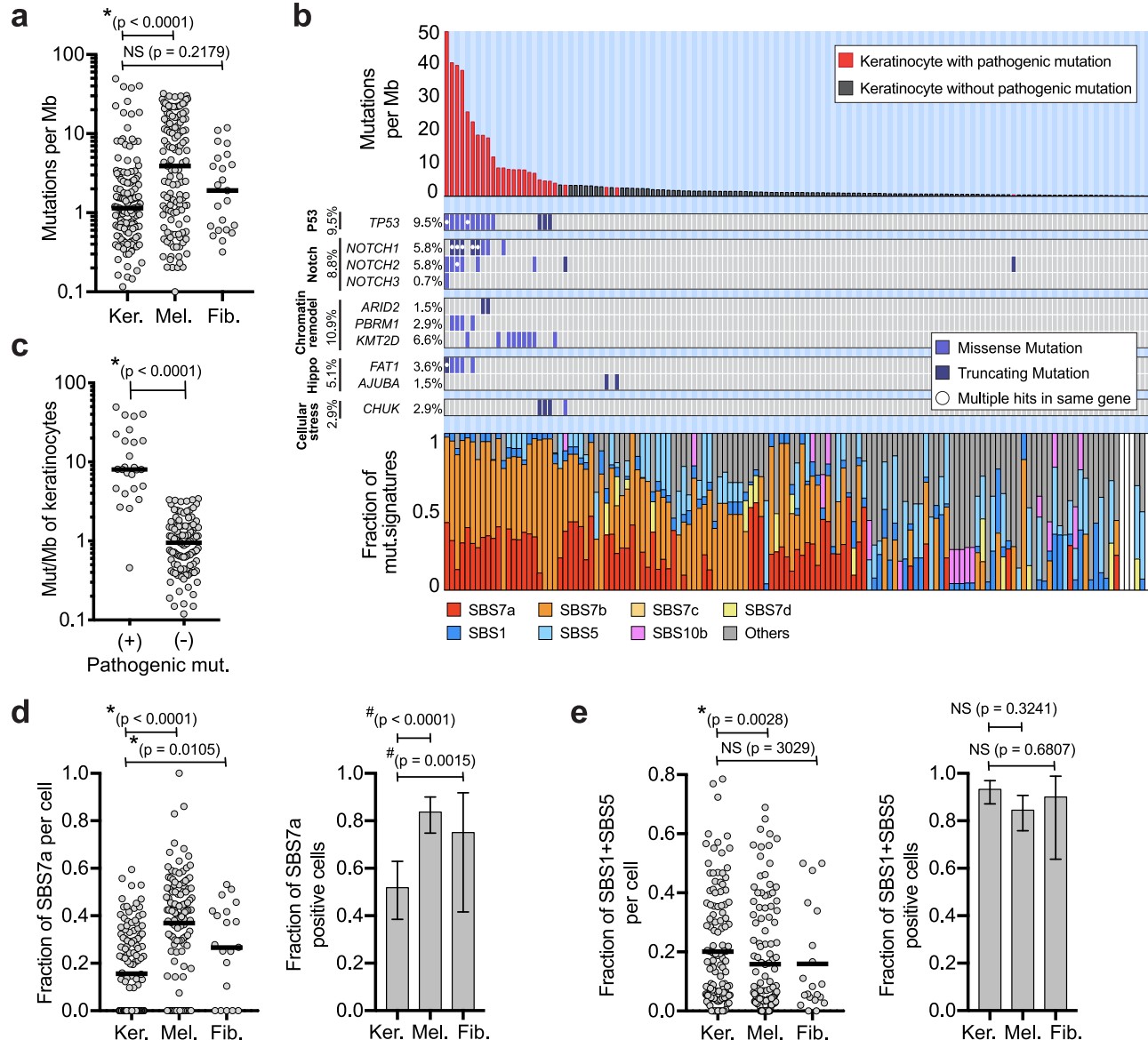

**Fig. 1 | Keratinocytes have distinct mutational landscapes compared to other cell types. a** Mutation burdens (mutations/megabase; Mut/Mb) of individual keratinocytes (Ker., *n* = 137) compared to melanocytes (Mel., *n* = 131) and fibroblasts (Fib., *n* = 23) derived from 22 independent skin biopsies from 15 unique donors. Each cell represents a single biological unit. **b** Mutation burden, driver mutations, and mutational signatures for 137 keratinocytes with each column of the three stacked panels representing an individual cell. Top panel: mutation burden of keratinocytes in descending order. Red bars indicate cells harboring one or more pathogenic mutations. Middle panel: tiling plot of pathogenic mutations (rows). Bottom panel: the fractions of different mutational signatures for each cell. White bars indicate keratinocytes with too few mutations to perform signature analysis. **c** Mutation burdens of keratinocytes with (*n* = 26) and without (*n* = 111) pathogenic mutations. **d** Left panel: fraction of mutations with UV signatures (SBS7a) in keratinocytes (*n* = 133), melanocytes (*n* = 123), or fibroblasts (*n* = 20). Right bar graph: fraction of cells with detectable SBS7a in keratinocytes (69/133), melanocytes (103/123), or fibroblasts (15/20). The center = fraction ± 95% confidence intervals (Poisson exact method). **e** The data is plotted as in (**d**) but for SBS1 and SBS5. Fractions of cells with SBS1 and SBS5: keratinocytes (124/133), melanocytes (105/123), and fibroblasts (18/20) ± 95% confidence intervals (Poisson exact method). All comparisons use single cells from separate donors as independent biological units. For all plots, an asterisk (*) or a hash (#) denotes *p* < 0.05 using the Wilcoxon rank-sum test (two-sided, cell to cell comparisons) or the Poisson test (two-sided, proportion comparisons), respectively. Horizontal bars show the median (panels a and c) or mean (panels d and e). Source data are provided as a Source Data file.

somewhat lower. The average mutation burden per keratinocyte in our study was 3.65 mutations/Mb. Other studies have inferred, from bulk-cell sequencing of epidermis, average mutation burdens per cell that ranged from less than 1 to greater than 20 Mut/Mb, though most report mutation burdens of approximately 5 mutations/Mb[10–12]. However, these measurements are not directly comparable because the data originated from different donors and body sites. Therefore, we performed bulk-cell sequencing of two epidermal microbiopsies, immediately adjacent to skin samples used for single-cell expansions (Fig. S6). In this patient-matched

comparison, the average mutation burden per cell from single-cell measurements was approximately three-fold lower than the mutation burdens inferred from patient-matched bulk-cell data. This difference may be caused by a random sampling, given that individual cells display a wide range of mutation burdens. As another possibility, the epidermis contains other cell types, including melanocytes, that may skew its mutation burden higher in bulk-cell measurements. However, it is possible that keratinocytes with low mutation burdens are more likely to clonally expand. Even if this latter explanation were true, our single-cell

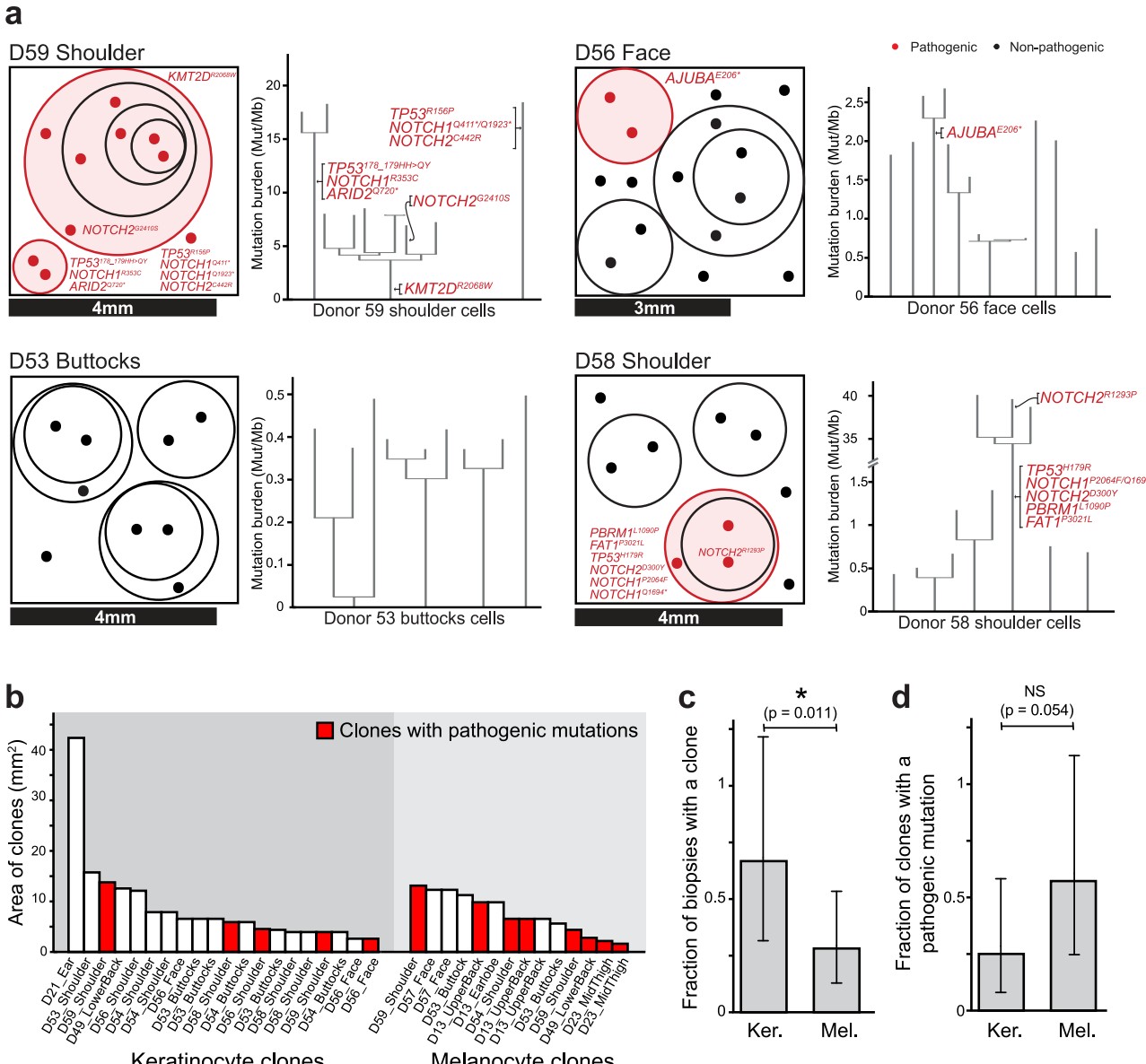

**Fig. 2 | Clonal architecture of keratinocytes in human skin. a** Clonal structure of keratinocytes from four representative skin biopsies (see Fig. S7 for all biopsies). Each biopsy represents an independent biological unit. The surface area of each biopsy is drawn to scale, as indicated, with dots representing the cells genotyped from each biopsy. The circles group phylogenetically related cells, with pathogenic mutations labeled in red. To the right of each schema, the corresponding phylogenetic trees, rooted in the germline state, are shown for all cells from that biopsy. **b** The area occupied by individual clones was calculated from the size of each biopsy and the proportion of cells attributed to each clone. Clone areas are shown for keratinocytes and melanocytes with clones harboring pathogenic mutations indicated in red. **c** Fraction of biopsies with a detectable clone of keratinocytes (Ker., 10 out of 15 biopsies) or melanocytes (Mel., 9 out of 32 biopsies) ± 95% confidence intervals (Poisson exact method). Here, Biopsies are treated as independent biological units. **d** Fraction of clones with an underlying pathogenic mutation in keratinocytes (5 out of 20 clones from 15 biopsies) or melanocytes (8 out of 14 clones from 32 biopsies) ± 95% confidence intervals (Poisson exact method). An asterisk (*) denotes $p < 0.05$ (two-sided Poisson exact test). Some donors contributed more than one biopsy from different anatomical sites. Source data are provided as a Source Data file.

genotyping workflow clearly reveals cell-to-cell mutation variation, which cannot be appreciated from bulk-cell sequencing.

Next, we analyzed skin cells for somatic copy number changes. Autosomal copy number alterations were infrequent, occurring in only 5.8% of keratinocytes, 13.7% of melanocytes, and 0% of fibroblasts. When copy number alterations were present, they typically affected a small portion of the genome (e.g., a single chromosomal arm), indicating that chromosomal instability is not a major mutational mechanism operating in normal skin cells.

Some keratinocytes shared a portion of their somatic mutations with other keratinocytes from the same biopsy, indicating that they are clonally related (Fig. 2a, S7). We inferred the area occupied by clones from the size of each biopsy and the proportion of cells with shared mutations. The median keratinocyte clone occupied 6.21 mm$^2$ (Fig. 2b). These surface area estimates are consistent with the upper end of clone sizes estimated by Martincorena and colleagues[10], who made their inferences from deep sequencing of bulk tissue. Our approach to clone detection is likely missing smaller clones, whose detection would require sequencing more cells per square millimeter. We compared biopsies in which keratinocytes and melanocytes were sampled at a similar density. Keratinocyte clones were more prevalent than melanocyte clones and less likely to harbor pathogenic mutations (Fig. 2c,

d). Interestingly, clones of keratinocytes with pathogenic mutations were not larger than clones of keratinocytes without pathogenic mutations (Fig. 2b). Martincorena and colleagues also found that clones with mutations in *NOTCH1*, *TP53*, or *FAT1* were only marginally larger than clones without pathogenic mutations[10].

### Genetic alterations driving the transition from actinic keratosis to squamous cell carcinoma

The accumulation of mutational damage can induce a keratinocyte clone to grow into a neoplasm known as an actinic keratosis, which has the capacity to further progress to squamous cell carcinoma. To better understand these transitions, we studied a cohort of archival tissues from 16 patients with squamous cell carcinoma, each immediately adjacent to an actinic keratosis (Supplementary Data S4, see Fig. 3a for an example). The histologically distinct regions were marked by a pathologist and dissected for DNA sequencing. Deep sequencing (380-fold coverage) was performed using a cancer gene panel (Supplementary Data S5). We prioritized sequencing depth over a broader sequencing footprint because keratinocyte cancers tend to have substantial levels of stromal cell contamination[6], and the high sequencing depth was helpful in resolving the different populations of clonally related cells within each dissection.

After DNA sequencing, somatic point mutations were stratified by their allele frequencies in each area to uncover the relationship between the dissected tissue regions. To our surprise, the squamous cell carcinoma was often not related to the neighboring actinic keratosis. In 6 of the 16 cases, the actinic keratosis and adjacent squamous cell carcinoma did not share somatic alterations (see Fig. S8a for an example), suggesting that the lesions arose as independent clones, despite their proximity. In 4 other cases, there were no mutations exclusive to the squamous cell carcinoma (see Fig. S8b for an example), implying there was a single population of clonally related cells spanning both dissected tissue areas, with no identifiable mutations accounting for the progression to invasive carcinoma. For these cases, the actinic keratosis histology may represent an extension of the squamous cell carcinoma rather than a distinct precursor lesion. The histopathological differences observed here may be attributed to differences in epigenetic mechanisms and the tumor microenvironment (TME), as seen in other cancers[34–37].

From the original 16 squamous cell carcinomas, there were only five that clearly evolved from the neighboring actinic keratosis, evidenced by having both a cluster of shared and unshared mutations, as indicated in Fig. 3b and S9. We prioritized these bona fide cases of squamous cell carcinoma arising from an actinic keratosis for further analyses, illustrated by an example case shown in Fig. 3.

We annotated mutations in genes known to drive keratinocyte cancers, and in the example case, the actinic keratosis had loss-of-function mutations affecting *TP53*, *NOTCH1*, *NOTCH2*, and *CDKN2A* as well as a gain-of-function mutation affecting the *TERT* promoter (Fig. 3c). The squamous cell carcinoma additionally acquired loss-of-function mutations in *ARID2* and *CBL*. ARID2 mutations disrupt the SWI/SNF chromatin remodeling complex[38]. The CBL mutation is a known RASopathy-associated variant that activates MAPK signaling[39]. There were also some mutations that clustered separately from the dominant clones of the actinic keratosis and squamous cell carcinoma (Fig. 3b, grey data points). These mutations had low allele frequencies in the actinic keratosis and/or squamous cell carcinoma. This could be due to subclones of cells, or more likely contamination from unrelated clones of keratinocytes in the tissue sample; therefore, we did not include these mutations in our phylogenetic analyses. There were no discernible copy number alterations in the actinic keratosis or squamous cell carcinoma of the example case (Fig. 3d), though there was allelic imbalance of chromosomal arm 9p, affecting the *CDKN2A* gene (Fig. 3e). Based on the distribution of shared and unshared somatic alterations in the dominant clones, we inferred the order in which

mutations occurred (Fig. 3f) and used immunohistochemistry to validate some of these observations. p53 immunoreactivity was present in both the actinic keratosis and squamous cell carcinoma (Fig. 3g), consistent with a missense mutation in *TP53* present in both areas. Higher phospho-MAPK signaling was observed in the squamous cell carcinoma (Fig. 3h), consistent with the *CBL* mutation in the squamous cell carcinoma, though it could be driven by other mutations and/or microenvironmental factors. Similar phylogenetic analyses were also performed on the other four squamous cell carcinomas that evolved from actinic keratoses (Fig. 4a).

To supplement our cohort, we reanalyzed publicly available data from another study that sequenced 160 known cancer genes in cutaneous squamous cell carcinomas and adjacent skin[18]. In that study, the adjacent skin biopsies were either: sun exposed skin, actinic keratosis, or squamous cell carcinoma in situ. We observed patterns similar to our cohort in this data. In most cases (Fig. S10), the squamous cell carcinoma was unrelated to any clones in the adjacent skin. In other cases (Fig. S11a), the squamous cell carcinoma likely extended into the neighboring skin. Finally, there were 3 bona fide cases in which the squamous cell carcinomas evolved from neighboring precursor lesions (Fig. S11b).

We explored common patterns of evolution in the 8 squamous cell carcinomas that clearly evolved from precursor lesions (5 from our cohort [Figs. 3f and 4a] and 3 from Kim and colleagues [Fig. S11b]). As expected, mutations in the p53 and NOTCH signaling pathways typically resided on the trunks of phylogenetic trees (Fig. 4b). Mutations that abrogate cell-cycle control checkpoints or upregulate telomerase also fell on the trunks of phylogenetic trees, indicating that these alterations contribute to the formation of actinic keratoses. By contrast, mutations that disrupt the SWI/SNF chromatin remodeling complex and mutations that activate the RAS/MAPK/PI3K-signaling cascade were most commonly observed at the transition to squamous cell carcinoma (Fig. 4b).

To complement the analyses of squamous cell carcinomas and matched precursor lesions, we also compared the frequency of driver mutations in publicly available data from unmatched cohorts of fully-evolved squamous cell carcinomas[6] and biopsies of normal skin[10] (Fig. 4c). *TP53* and *NOTCH* mutations were common in normal skin, confirming that they undergo selection early, even before neoplasms are present. *TP53* mutations were less common in normal skin than *NOTCH1* mutations but more common in squamous cell carcinoma, suggesting that *TP53* mutations endow keratinocytes with more malignant potential, as has previously been shown in the esophagus[40,41]. Mutations in other genes, such as *CDKN2A* and *ARID2*, were rare in normal skin but common in squamous cell carcinoma, implying that they undergo selection comparatively later in tumor evolution. A limitation to these comparisons is that the normal skin biopsies were sequenced with a small gene panel, precluding a comprehensive comparison of mutation frequencies in all genomic loci, such as the *TERT* promoter.

### Spatial transcriptomic analysis of actinic keratoses adjacent to squamous cell carcinomas

Bulk-cell RNA-sequencing has been performed on normal skin, actinic keratoses, and squamous cell carcinomas[16,17,42], providing insights into the gene expression changes that occur during tumor evolution. However, in those studies, the data encompass mixtures of clones whose phylogenetic relationships are unknown. Here, we performed spatial transcriptomics (10X Visium) on five of the squamous cell carcinomas adjacent to actinic keratoses, whose clonal relationships were resolved. Compared to bulk RNA sequencing, spatial transcriptomics typically provides lower coverage and does not capture splicing patterns. However, for the purposes of this study, these limitations were outweighed by the added value of spatial information.

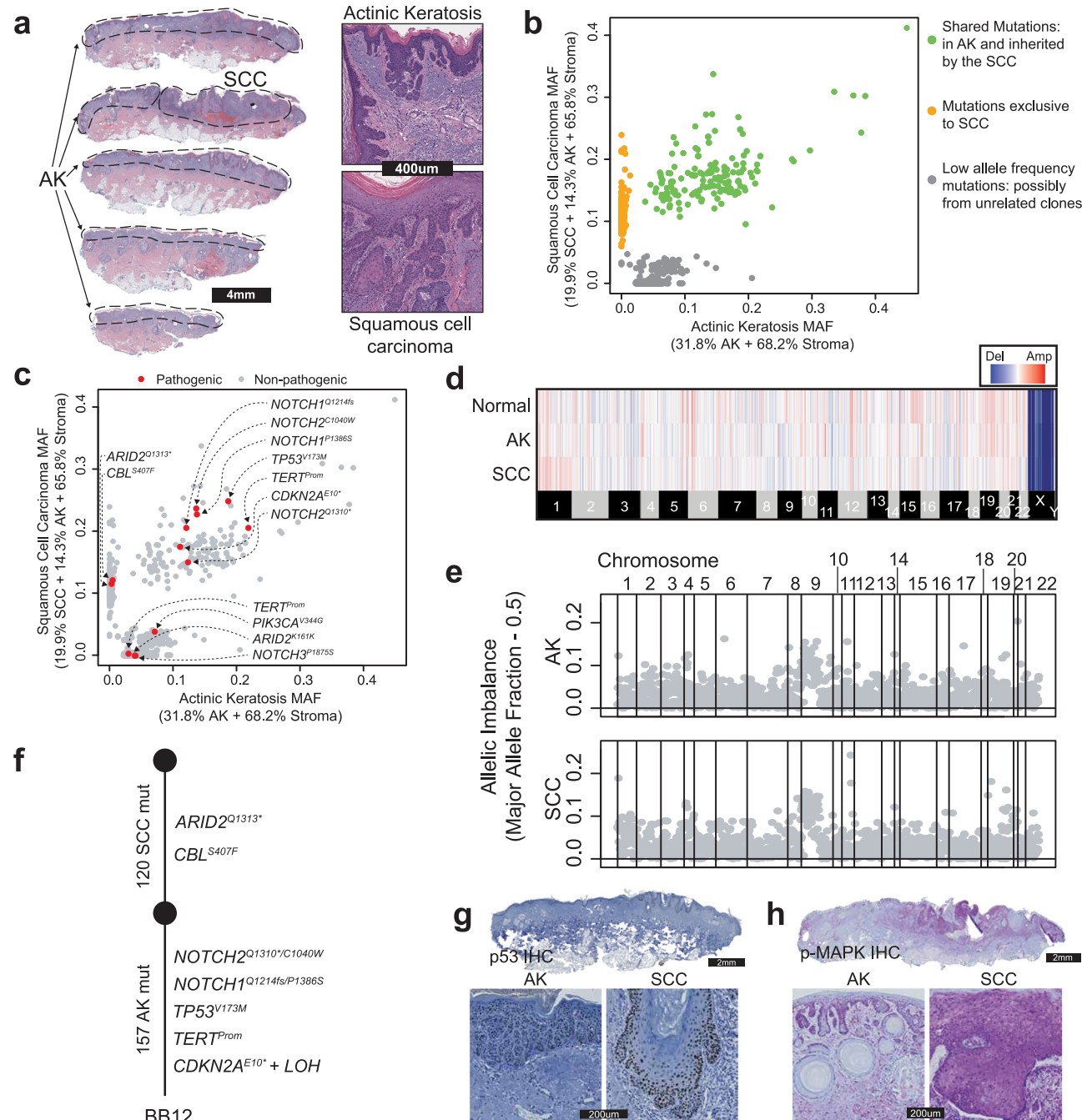

**Fig. 3 | The genetic evolution of a cutaneous squamous cell carcinoma from an actinic keratosis. a** H&E-stained section of a skin biopsy with adjacent areas of squamous cell carcinoma (SCC) and actinic keratosis (AK) dissected, as indicated by the dashed lines. Skin biopsies from 16 different donors (Supplementary Data S4) were analyzed similarly. In five independent cases, squamous cell carcinoma evolved from adjoining actinic keratosis, as shown in this figure (see Fig. S9 for all the cases). Representative images of cases where squamous cell carcinoma did not evolve from adjoining actinic keratosis are shown in Fig. S8. **b** Scatter plot of mutant allele fractions (MAF) in the squamous cell carcinoma and actinic keratosis reveal three clusters of mutations. **c** The same scatterplot as shown in panel b with pathogenic mutations annotated. **d** Copy number alterations were inferred over bins of the genome (columns) for each histologic area (rows) and are shown as a heatmap (red = gain, blue = loss, white = no change). No somatic gains or losses were observed. **e** Major allele frequency−0.5 (y-axis) for heterozygous SNPs across the genome (x-axis) show loss of heterozygosity over chromosome 9p. **f** Phylogenetic tree based on somatic mutations (mut) rooted at the germline state. **g, h** Representative images of immunohistochemistry (IHC) for p53 (**g**, brown stain) and phospho-MAPK (panel h, purple stain), show keratinocytes overexpressing p53 in both regions with increased phospho-MAPK (Mitogen-activated protein kinase) in the squamous cell carcinoma (*n* = 5, all the cases where squamous cell carcinoma evolved from adjoining actinic keratosis). Source data are provided as a Source Data file.

The spatial transcriptomics data helped define the localization of tumor cells, revealing a complex spatial architecture of clones. We inferred copy number from individual spots of the Visium arrays[43], and identified spots with copy number profiles similar to those observed in bulk-cell DNA-sequencing data (Fig. S12a). In each case, the main lesion of squamous cell carcinoma had concordant copy number alterations, but interestingly, the spatial data revealed satellite colonies of cells, physically distant from the main tumor (Fig. S12c). The presence of areas of squamous cell carcinoma outside the main lesion was consistent with DNA-sequencing data, where we recurrently inferred

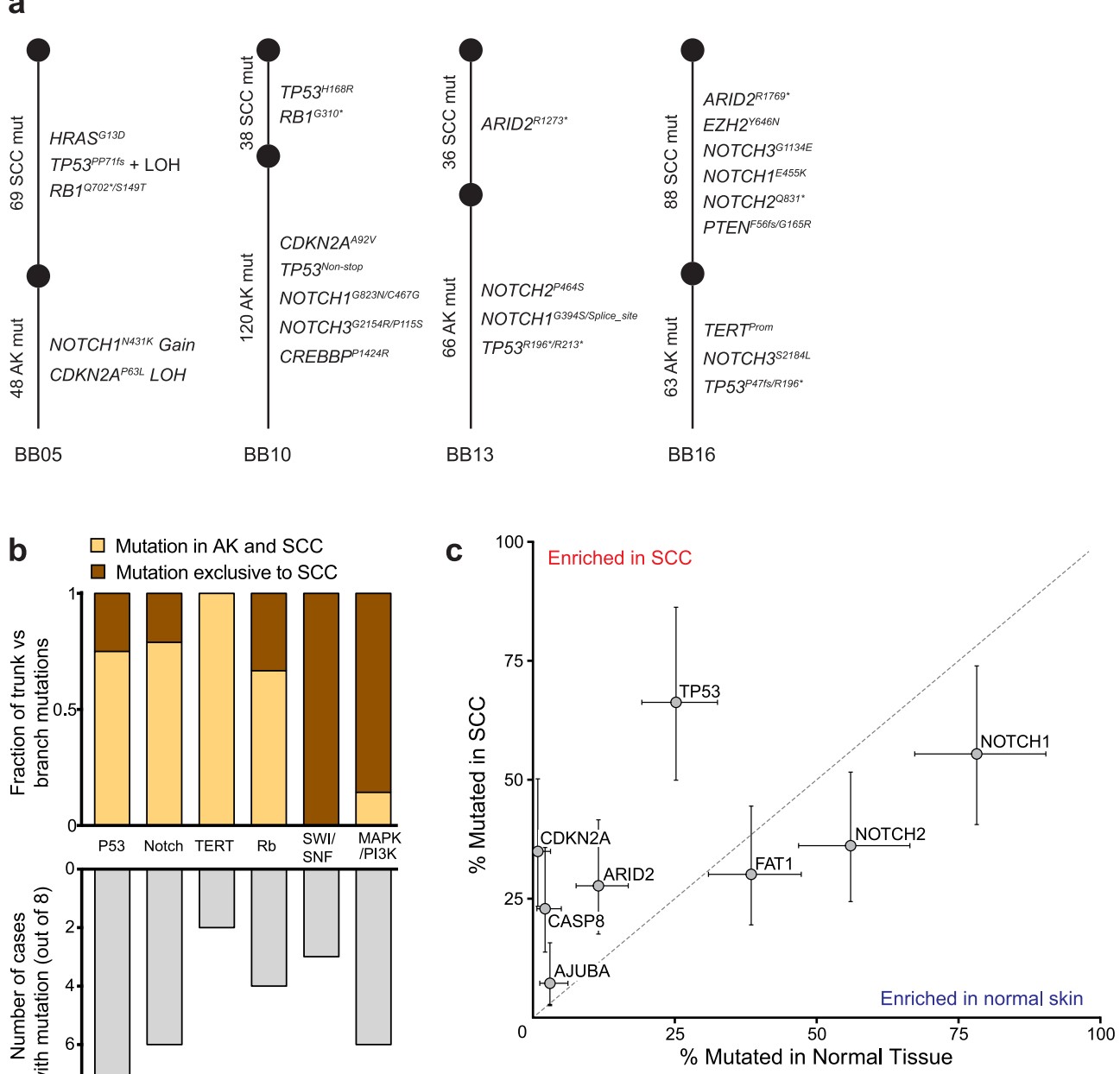

**Fig. 4 | The sequential order of genetic alterations during progression from actinic keratosis to squamous cell carcinoma. a** Phylogenetic trees based on somatic mutations (mut), rooted in the germline states, summarize the evolution of four squamous cell carcinomas (SCC) that evolved from actinic keratoses (AK). See figure S9 for further details on these four cases and Fig. 3 for a summary of the example case. **b** Eight squamous cell carcinomas that evolved from neighboring precursor lesions were identified as described. The stacked bar plot (top panel) indicates the proportion of mutations, recurrently mutated in these eight cases, in the trunk versus branch of phylogenetic trees. The bar plot (lower panel) indicates the number of cases with a mutation in each pathway. Mutations in the p53, Notch,

TERT, and Rb pathways tended to occur early, contributing to the formation of actinic keratoses. Mutations affecting the SWI/SNF chromatin remodeling complex or activating the MAPK/PI3K pathways tended to occur later, driving the transition to squamous cell carcinoma. **c** The scatterplot shows the frequency of mutations in selected driver genes in normal skin biopsies (total mutations = 234, x-axis) versus squamous cell carcinoma (total mutations = 83, y-axis). Horizontal and vertical error bars for each gene represent 95% confidence intervals for mutation frequencies in normal skin and squamous cell carcinoma, respectively (Poisson exact method). A y = x line is included for orientation. Source data are provided in the Source Data file.

low levels of cross-contamination between microdissected regions (Figs. 3b, S9).

Spots clustered primarily by cell type (Fig. S13b) and secondarily by cell state (Fig. S13c, d, S12b). We selected spots overlying actinic keratosis or squamous cell carcinoma, aided by the distribution of copy number alterations, and performed differential gene expression analyses. On average, spots overlying squamous cell carcinoma expressed higher levels of stem cell,

progenitor, and mesenchymal genes, in agreement with bulk-cell data[42] (Fig. S13c), but the spatial data revealed notable heterogeneity within these tumors.

Within actinic keratoses and squamous cell carcinomas, gene expression programs spanned a range of differentiation states (Fig. S13d, e). Spots with stem-like signatures were exclusive to the invasive front of squamous cell carcinoma, corresponding to the tumor-specific keratinocytes or TSKs, defined by Ji and

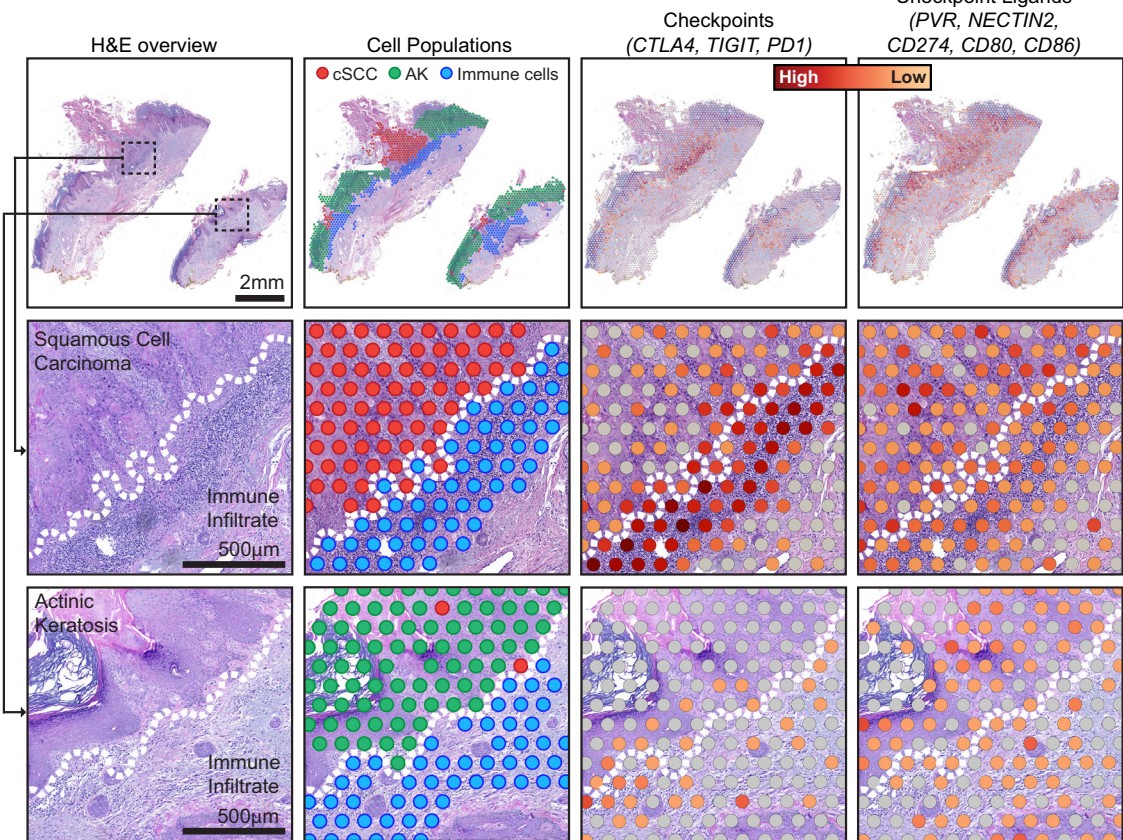

**Fig. 5 | Spatial heterogeneity in gene expression of immune cells at the interface of squamous cell carcinoma versus actinic keratosis.** Each column of images shows a different view of spatial transcriptomic data from a representative case BB05, including: an H&E overview, annotated spots, and gene expression of immune checkpoints and their ligands. Gene expression intensities represent the combined expression of the checkpoint or ligand genes listed. Zoomed insets show the interface of tumor epithelia and immune cells, illustrating different levels of checkpoint and ligand expression in squamous cell carcinoma versus actinic keratosis. Dotted lines indicate the tumor/immune boundary. This analysis was repeated independently in five cases, and the images shown are representative. See Fig. S14 for an overview of other cases.

colleagues[44]. Cells within both actinic keratoses and the deeper regions of cutaneous squamous cell carcinomas expressed gene programs associated with basal, suprabasal, spinous, and corneocyte differentiation. This indicates that tumors retain a broad spectrum of epithelial cell states. Notably, even as the average transcriptional state of epithelial cells shifts toward a progenitor-like fate during tumor progression, hierarchical differentiation programs remain preserved, even in fully developed tumors. A limitation of these analyses is that the squamous cell carcinomas profiled in this study were all histopathologically classified as well-differentiated. Future studies incorporating tumors with a broader range of differentiation phenotypes will be needed to better characterize intratumoral heterogeneity.

Finally, we observed spatial heterogeneity in gene expression of non-tumor cells. It was common for immune infiltrates to extend along the borders of both the actinic keratosis and squamous cell carcinoma, but immune cells expressed different gene expression programs, depending on their localization (Figs. 5, S14). We observed higher expression of immune checkpoint ligands (*PVR, NECTIN2, CD274, CD80,* and *CD86*) in the tumor cells at the invasive front of the squamous cell carcinomas. Concordantly, we observed higher expression of immune checkpoint proteins (*CTLA4, TIGIT,* and *PDCD1*) in the lymphocytes at the invasive front of squamous cell carcinomas. To elucidate how immune phenotypes evolve during progression, future studies will need to profile additional tumors with confirmed phylogenetic relationships between squamous cell carcinomas and their adjacent precursor lesions.

## Discussion

Our work provides different vantage points into the changes that occur during the transformation of keratinocytes to squamous cell carcinoma (Fig. 6), starting with individual keratinocytes of normal epidermis. Our single-cell analysis revealed a surprisingly broad range of cellular mutation burdens. Cells with high mutation burdens harbored pathogenic mutations, typically affecting *TP53* or *NOTCH1*. p53 conveys DNA damage signals and induces cell cycle arrest to increase repair time or cell death when damage surpasses a threshold, such as after a sunburn[9]. Cells with defective p53 are thus likely to accumulate DNA damage at a higher rate. *NOTCH1* mutations induce a stem/progenitor cell state in epithelial cells[45], which may also prevent cells from undergoing apoptosis upon excessive DNA damage. Loss-of-function mutations in *TP53* and *NOTCH1* are known to provide a fitness advantage to epithelial cells[40,46,47], but our findings suggest their contribution to tumor progression may primarily stem from the mutator phenotypes that they induce. Indeed, *TP53* and *NOTCH1* mutant clones were no larger than clones without pathogenic mutations.

*TP53* and *NOTCH1* mutations thus likely prime keratinocytes for transformation by increasing their mutation rates, but additional driver mutations are needed to form a neoplasm (Fig. 6). To uncover the secondary mutations and deduce the order in which they undergo selection, we compared the mutational landscapes of normal epidermis, actinic keratoses, and squamous cell carcinomas. *FAT1* mutations, which inactivate the Hippo pathway and thereby promote the downstream oncogenic activity of YAP/TAZ transcriptional coactivators, were observed in individual keratinocytes of normal epidermis,

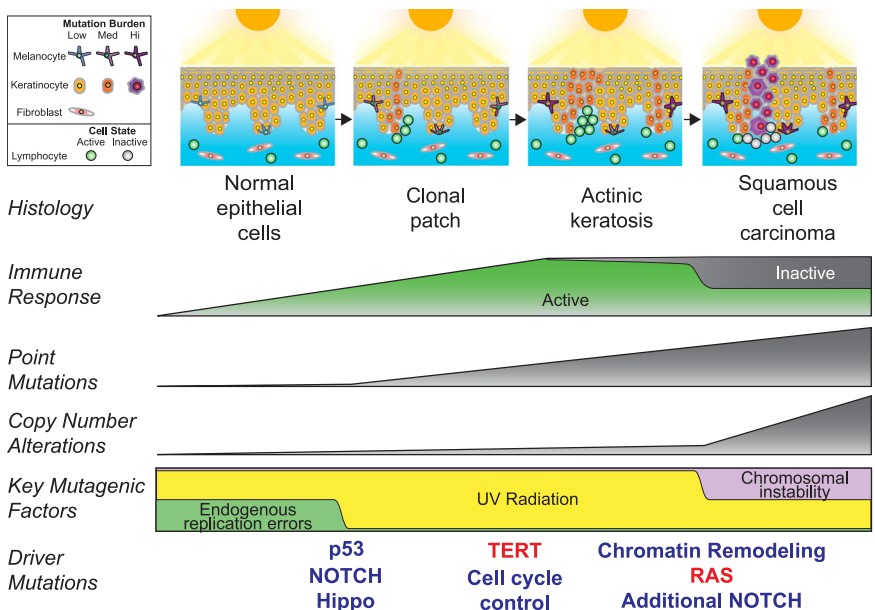

**Fig. 6 | Summary of key events that occur during the evolution of cutaneous squamous cell carcinoma.** After continual exposures to UV radiation, fibroblasts modestly increase their mutation burdens, melanocytes sharply increase their mutation burdens, and keratinocytes have a mixed response. Most keratinocytes accumulate little mutational damage, but a subset with pathogenic mutations builds up mutations more rapidly than other skin cells. UV radiation induces expansion of independent clones of keratinocytes, often in close proximity and admixed, resulting in a complex clonal structure whereby adjacent lesions are not necessarily related. Driver mutations undergo selection in a stereotypical order, linked to histologic and genetic changes that occur during tumor evolution. An immune response builds during progression, but activity is blunted via engagement of immune checkpoints in squamous cell carcinoma.

though they were less common than *TP53* and *NOTCH1* mutations. Loss-of-function mutations of *CDKN2A* and *TERT* promoter mutations were rare in normal epidermal cells but were recurrently present in actinic keratoses. Finally, *ARID2* mutations, which disrupt the SWI/SNF chromatin remodeling complex, and gain-of-function mutations in the RTK-RAS-MAPK pathway were enriched specifically in squamous cell carcinomas. More work will be needed to understand how the disruption of SWI/SNF and activation of RTK-RAS-MAPK drive the transition from precancer to cancer in the setting of keratinocytes. Disruption of SWI/SNF may help keratinocytic neoplasms evade immune recognition[48], though the mechanisms by which SWI/SNF loss promotes tumorigenesis in any cancer remains an open question[49]. Activation of RTK-RAS-MAPK signaling promotes growth and survival in cancer cells[50], which may drive the transition of an actinic keratosis to squamous cell carcinoma.

The spatial architecture of keratinocyte clones in tumor-bearing skin revealed a complex mosaic of neoplastic and non-neoplastic cells. Squamous cell carcinomas were often genetically unrelated to neighboring precursor lesions, suggesting that collisions of clonally unrelated neoplastic proliferations of keratinocytes are common. When we inferred the localization of clones in spatial transcriptomic data, the boundaries of keratinocyte clones were not contiguous in 2-dimensional space. Lineage-tracing experiments in mice show that clones of epithelial cells from surrounding tissue can invade, infiltrate, and mix with tumors as they grow[51,52], which may explain the complex patchwork of clones we observed. For clinical purposes, it is not safe to assume that keratinocyte lesions in close proximity are necessarily clonally related or that the histological boundaries are reliable measures of tumor extent.

Finally, we observed a remarkable degree of spatial heterogeneity in gene expression. A stratified hierarchy of differentiation, resembling that of normal epidermis, was maintained within actinic keratoses and squamous cell carcinomas, though cells from squamous cell carcinomas were, on average, less differentiated. The composition of immune cell infiltrates also varied within tumors. The immune cells near the leading edge of squamous cell carcinomas expressed high levels of immune checkpoint genes. These observations suggest that an immune response is mounted already at the actinic keratosis stage, possibly resulting in an equilibrium state of oncogene-mediated proliferation and immune cell-mediated elimination of partially transformed keratinocytes. However, that state is broken during tumor progression once cells of the carcinoma develop the ability to engage immune-cell checkpoints. It is unclear whether this immune evasion is driven by specific somatic mutations in tumor cells, but it is intriguing that mutations affecting the SWI/SNF chromatin remodeling complex are common at the transition from actinic keratosis to squamous cell carcinoma because there is an emerging view that these mutations drive immune evasion[48].

In closing, our study reveals key events during the transformation of keratinocytes to squamous cell carcinoma. While validation studies are still needed, the genetic alterations identified in this study represent promising candidate biomarkers for improving the diagnosis and prognosis of keratinocyte carcinomas. Squamous cell carcinomas respond favorably to immune checkpoint inhibitors, likely due to their high mutation burdens. Given the widespread presence of TP53-mutant keratinocytes with high mutation burdens in clinically normal skin, future studies should explore whether immunotherapies can also target and eliminate precancerous cells as a potential prophylactic strategy.

## Methods

### Collection of normal human skin samples for keratinocyte genotyping

Skin biopsies were collected from cadaver tissue through the University of California, San Francisco Willed Body program or from patients seen by Dermatologists at the University of California, San Francisco or Northwestern University. Living patients consented to participating in this study through approved protocols by the Institutional Review Board of the University of California, San Francisco (22-36678) and the Institutional Review Board of Northwestern University (STU00211546). Cadaver tissue came from donors who broadly consented, prior to their death, as part of their living will, to the use of

their tissues for medical research and/or educational purposes. Donor age, sex, and gender were recorded from their preapproved Vital Statistics Information Sheet. The living donors provided self-reported information and consent, in accordance with their IRBs, to publish data on their UV radiation exposure and various risk factors to skin cancer, including age, sex, ancestry, sun exposure, sunscreen use, and tanning bed use, by completing a questionnaire. For donors from the Willed Body Program, similar information was obtained through their consented Vital Statistics Information sheets. The detailed information on the risk factors is listed in Supplementary Data S2. In both instances, we typically took small shave biopsies (3–5 mm in their longest dimension) of skin samples. Due to the small size of the biopsies, the entirety of each tissue was utilized for single-cell analysis, preventing histological assessment of features such as the degree of solar elastosis. As a result, we relied on available data, including self-reported information, to estimate cumulative sun damage. The absence of histological evaluation represents a limitation in precisely quantifying cumulative sun exposure.

In addition to skin samples, which were used for single-cell genotyping, we collected a source of reference DNA from each donor. Living donors submitted a buccal swab, which was used as a reference source of germline DNA (Supplementary Data S2). Reference DNA from cadaver tissue came from a separate tissue biopsy, unrelated to and distant from the skin sample being processed for somatic mutation analyses. Samples from both male and female donors were included in all analyses to minimize sex-related biases in the results.

## Sequencing DNA/RNA from individual skin cells

Skin cells were genotyped[23]. Due to the small size of the skin samples, this procedure consumes the entire specimen. Below, we overview of the genotyping workflow with an emphasis on modifications made in the present study.

Each skin biopsy was treated overnight in dispase (10 mg/ml) to break up collagens holding the dermal layer of the skin to the epidermis. After treatment, the epidermis was separated from the dermis with tweezers, minced, and trypsinized to form a suspension of single cells. The single-cell suspension of epidermal cells was divided into two portions, and cells from each portion were cultured under different conditions. One portion of cells was plated in CNT40 media (CELLnTEC), which favors melanocyte proliferation, and the other portion of cells was seeded in KSFM media (Gibco, 10724-001), which favors keratinocyte proliferation. The remaining dermis was separately minced, broken down by 0.2 mg/ml collagenase (Roche) at 37 °C for 30 minutes, and filtered through a 40 µm nylon mesh. The resulting single-cell suspension of dermal cells was seeded in DMEM (Gibco, 11965092), favoring the outgrowth of fibroblasts.

The bulk cultures of cells were allowed to recover until they showed signs of stabilization and proliferation (typically 2–10 days). After their establishment, the bulk cells were manually single-cell sorted into individual wells of a 96-well plate using limited dilution. The cells were diluted so that, on average, one cell would be deposited in every other well (concentration of 0.5 cells/well). We chose limited dilution over the usage of flow cytometry because it yielded a higher proportion of surviving cells. When using the limited dilution method, a healthy population of bulk cells would successfully seed new plates at approximately 50% efficiency, suggesting that the seeded cells were representative; however, we cannot entirely rule out the possibility of a bias being introduced by this step.

Prior to single-cell sorting, the bulk cultures of cells were primarily composed of either fibroblasts, melanocytes, or keratinocytes, depending on the media in which they were maintained, but each culture, nevertheless, contained a mixture of cell types. To further enrich for melanocytes or keratinocytes, we performed differential trypsinization when sorting individual cells from bulk-cell cultures of epidermis. Melanocytes are less adherent than keratinocytes and can

be separated with a quick (typically 3 minutes) treatment of trypsin (0.05%), whereas keratinocytes require a longer or sometimes multiple treatments to be released from the tissue culture dish. After single-cell sorting, each cell type was maintained in its optimal media and allowed to clonally expand.

Plates were screened a day after seeding, and wells with more than 1 cell were not processed further to ensure that colonies started from a single cell. As an additional safeguard, we discarded data from colonies with mutant allele frequencies averaging less than 50%, revealed at the analysis step, indicating that these wells had more than one founder cell.

Cells were clonally expanded over a period of 2–4 weeks, typically forming colonies of approximately 200 cells before undergoing growth arrest. It is unclear why cells arrest after a brief period of growth, but we believe the daughter cells maintain a memory of mitotic signals, at least for a few population doublings, as previously described[53], before stress signals build up and induce growth arrest. While the increase in cell number was modest, it relieved a major bottleneck that ordinarily prohibits the detection of somatic mutations at high specificity and sensitivity from a single cell.

We considered the possibility that the brief period of tissue culture would introduce somatic mutations. However, we[23] and others[27] showed that the number of mutations from 2–10 days in tissue culture (the period of time in which the cells were maintained in culture prior to single-cell seeding) was minimal (at most, 0.1 mutations/Mb). Of note, somatic mutations that arose after single-cell seeding and during clonal expansion had subclonal allele frequencies and were removed.

Clonal expansion increased the starting template material, but the genomic material from 200 cells is insufficient to be directly sequenced with standard next-generation sequencing workflows. Therefore, we extracted and separated genomic DNA and mRNA using the G&T-seq protocol[54,55]. The mRNA was amplified with a modified version of the SMART-seq2 protocol, as described[54,55]. The genomic DNA was amplified with multiple displacement amplification (Qiagen REPLI-g Single Cell Kit, 150345) or via primary template amplification[56] (BioSkryb ResolveDNA Whole Genome Amplification Kit, 100136).

Amplified cDNA, amplified genomic DNA, or bulk-cell genomic DNA (reference tissue) samples were prepared for next-generation sequencing. Nucleic acids were sheared to a target size of 350 bp (Covaris LE220), end-repaired, ligated to IDT 8 or 10 dual index adaptors, and amplified using KAPA HyperPrep Kit (Roche, KK8504). Libraries were enriched for exomic sequences by hybridization with NimbleGen SeqCap EZ Exome + UTR (Roche, 06740294001) or KAPA HyperExome (Roche, 09062556001) baits, according to the manufacturer's protocols. Paired-end sequencing (either 100 or 150 bp) was performed on one of the following Illumina instruments: Illumina HiSeq 2500 or NovaSeq 6000.

DNA-sequencing data were aligned to the hg19 version of the genome with BWA (v2.0.5)[57]. Sequencing reads were deduplicated with Picard (v2.1.1) and further curated (indel realignment and base quality recalibration) with GATK (v4.1.2.0). RNA-sequencing data was aligned to the genome and transcriptome with STAR align (v2.1.0)[58] and deduplicated with Picard (v2.1.1). Gene-level read counts were quantified with RSEM (v1.2.0)[59].

## Verifying the genetic ancestry of donors (Related to Fig. S2 and Supplementary Data S2)

We first downloaded Phase 3 autosomal variant data from the 1000 Genomes Project (ftp://ftp.1000genomes.ebi.ac.uk/vol1/ftp/release/20130502; PED file: ftp://ftp.1000genomes.ebi.ac.uk/vol1/ftp/technical/working/20130606_sample_info/20130606_g1k.ped) and removed multi-allelic variants, indels, and SNPs with allele frequencies <1%. For the remaining loci, BCFtools and SAMtools[60] were used to call genotypes from the bulk exome-seq data of the donors. Loci with fewer than 10 mapped reads in any individual were considered as

having unknown genotypes and excluded. Loci with alternative alleles not represented in the 1000 Genomes Project were also removed. The data from both our cohort and the 1000 Genomes Project were then converted to PLINK format and merged using PLINK v1.90b6.24[61]. From the merged dataset, the SNPs with allele frequencies <5% were removed. The dataset was further pruned to retain variants in approximate linkage equilibrium using the parameters [−-maf 0.05 −indep 50 5 1.5]. The final set of 44,689 SNPs was used for principal component analysis (PCA). The R library scatterplot3d[62] was used for visualization.

### Confirming the lineage of the cell type (Related to Fig. S1c, d)
In addition to using morphology to identify the lineage of each cell, we inferred cell identity based on gene expression patterns. A t-SNE plot was generated using the Rtsne R package (v0.16), showing three distinct clusters of cells. Morphologically, the three clusters of cells corresponded to keratinocytes, melanocytes, and fibroblasts. We performed differential gene expression analysis to identify the top genes associated with each cluster using DESeq2 R package (v1.38.3)[63]. The top genes are shown in Fig. S1 as a heatmap.

### Point mutation calling from single-cell expansions
Somatic mutations were called from colonies of individual skin cells[23] and summarized below. Code related to these operations can be found here: https://github.com/ShainLab.

MuTect2 (v4.1.2.0) was used to generate a candidate list of point mutations by comparing the aligned bam files of each single-cell expansion to the BAM files representing the respective patient's normal DNA. Pindel was used to generate a candidate list of short insertions and deletions using the same comparison. Pindel calls were filtered to identify candidate mutations with at least 4 reads of support, which were manually inspected to eliminate alignment artifacts. These steps removed sequencing- and alignment-induced artifacts but not artifacts induced during template amplification.

We used two strategies to distinguish bona fide mutations from amplification-induced artifacts. First, mutations that were present in both the DNA-sequencing and RNA-sequencing data were considered to be true mutations because it is unlikely the same artifact would be introduced when amplifying DNA and also when amplifying RNA. Mutations present in DNA-sequencing data but absent in RNA-sequencing data of at least 15X coverage were considered to be artifacts with some exceptions. An exception was made for truncating mutations (i.e., nonsense, splice-site, or frameshift mutations) because a truncating mutation, encoded in DNA, is likely to undergo nonsense-mediated decay after expression and may not be detectable in RNA-sequencing data. An exception was also made for X-chromosome mutations from female samples because mutations on the silenced X chromosome may not be expressed.

Second, we considered a variant to be a true mutation if it occurred in complete linkage with one of the alleles from a nearby heterozygous SNP – a strategy that has also been validated and utilized by others[64]. Mutations that were not in complete linkage with nearby SNPs were considered to be artifacts unless they occurred in a region with copy number gains. This strategy works well because we genotyped colonies of cells (as opposed to individual cells). Since there were multiple template molecules, corresponding to each allele, an artifact rarely appears in complete linkage with either haplotype after amplification.

The strategies, described above, enabled us to validate (or invalidate) variants within expressed genes and/or variants that could be phased into their respective haplotypes. We used the variant allele frequencies of these credentialed variants to establish a benchmark to determine the statistical likelihood that the remaining variants (those in poorly expressed genes and unphased portions of the genome) were bona fide mutations or artifacts. Most artifacts had low allele frequencies, whereas bona fide mutations tended to have allele frequencies of 50% (for heterozygous mutations) or 100% (for homo- or hemi-zygous mutations). See Tang et al. for more details[23] on how we arrived at a specific cutoff for each sample.

### Copy number calling from single-cell expansions
Copy number was inferred from colonies derived from individual skin cells using CNVkit (v.0.9.6.2). Our copy number workflow is described in detail here[23]. Briefly, CNVkit infers copy number from either DNA-sequencing[65] or RNA-sequencing[66] data. Since we produced matching DNA/RNA-sequencing data from each colony, we ran CNVkit in both modes. When running CNVkit on either DNA- or RNA- sequencing data, we generated a reference from large pools of samples that were of the same lineage and run in the same sequencing batch. We considered a copy number alteration to be true when it was detected in both the DNA- and RNA- sequencing data. Copy number inferences at the bin level (.cnr files) or segment level (.cns files) are available here: https://figshare.com/s/9474ef6f59d92dc082f8.

### Allelic dropout from single-cell expansions
A set of germline heterozygous SNPs was identified from the reference BAM of each donor as described[23]. Briefly, we called variants in the reference BAM against the reference hg19 genome using FreeBayes (v.1.3.1), identified variants that had been observed in greater than 1% of participants from the 1000 Genomes Project, and required variants to have at least 5 reads supporting each allele and a variant allele frequency between 40-60% for each allele.

After establishing these sets of germline heterozygous SNPs for each donor, the number of reference and alternate reads was counted in the BAM files from single-cell expansions of each donor. We calculated rates of mono-allelic and bi-allelic dropout for each colony. We also consulted the copy number data to distinguish between biological dropouts, resulting from a deletion, versus technical dropouts, resulting from amplification biases during sample preparation. Dropout rates are listed in Supplementary Data S1 in columns L and M. The median allelic dropout rate across all single-cell expansions, was 0.06%, confirming our ability to sensitively detect single-nucleotide variants in single-cell expansions.

### Mutation burden and signature analysis (related to Fig. 1, S3, and S5)
Mutation burdens were calculated as mutations per megabase. The number of mutations was tabulated for each cell and divided by the captured footprint that was sequenced with 10X coverage or greater. The footprint of sequencing data with 10X coverage or greater was counted with the footprints software[23]. A trinucleotide profile for each individual cell was generated using deconstructSigs R package (v1.9.0)[67]. The Bioconductor library BSgenome.Hsapiens.UCSC.hg19 (v1.4.3) was first used to apply mutational context to all the single-base substitutions (SBS) mutations identified in each cell. The results for all the cells were combined for each cell type and visualized as trinucleotide context in Fig. S5. A custom forward stagewise algorithm using SigProfilerAssignment (v0.1.8) was applied to build a mutational profile based on 78 pre-defined COSMIC (v3.7) signatures previously extracted by SigProfiler[68]. The minimum number of SBS mutations for the signature analysis is set at 10. The signatures for all the cells are depicted as stacked barplots in Fig. 1b (bottom panel), showing the fractions of top 7 signatures. Signatures present in less than 10% of cells were grouped into an [others] category.

### Annotation of pathogenic mutations in individual cells (related to Fig. 1b and S3b)
As a guide for annotation of pathogenic mutations, we identified mutations in genes shown to be under selection from a meta-analysis of cutaneous squamous cell carcinoma[6]. Three authors on this

manuscript independently reviewed the mutation lists to nominate pathogenic mutations. After independent review, we consulted and agreed upon a single list, as annotated in column AA of Supplementary Data S3. The full list of mutations, including passenger mutations, is also available in Supplementary Data S3 for interpretation by the readers.

### Construction of phylogenetic trees from individual skin cells (Related to Fig. 2a)

After calling mutations in individual cells, overlapping mutations between cells from the same donor were identified. Only mutations with at least 10X coverage were considered for phylogenetic analyses – we made this decision to reduce the risk of calling a mutation private to one cell when coverage over the mutant site was low in other cells. If any mutations were shared between 2 or more cells, we ran the mpileup function of SAMtools to count the reference/mutant reads for all other mutations in all cells to ensure that they were genuinely private to each of the cells. Only in rare cases, these identified mutations that were in fact shared but had been missed by the mutation calling algorithm because they were just below the MuTect2 detection thresholds. In these rare cases, the mutation was added to any samples for which it was missed.

After compiling a list of shared and private somatic mutations across the cells, a phylogenetic tree was constructed using an R script that employs the dplyr, tidyr, and ggplot2 packages. The script is publicly available for use at [https://github.com/delahny/phylogenetictrees]. It identifies and counts both shared and private mutations to generate datasets that correspond to branches and trunks, which are plotted by applying hierarchical leveling. The resulting phylogenetic tree was rooted at the germline state, with trunks scaled according to the number of shared mutations and branches scaled based on the number of private mutations. Each cell's identity was labeled at the terminus of its branch, and cells without a vertical branch had no private mutations.

### Construction of clonality plots (Related to Fig. 2a and S7)

After constructing phylogenetic trees, we drew schematic images to depict the clonal architecture of each biopsy. In these plots, individual cells are represented as points, with phylogenetically related cells grouped within circles. Cells sharing pathogenic mutations and their corresponding clones are highlighted in red. In these schematics, the precise spatial localization of a given cell is unknown.

To estimate the total surface area of each biopsy, we calculated the surface area based on the diameters of the biopsies. For biopsies with non-circular shapes, measuring scales present within the biopsy images were used to calculate the area with ImageJ. For each set of clonally related cells, we estimated the surface area using the formula: (number of phylogenetically related cells / total cells genotyped from the biopsy) * total area of the biopsy. In samples with subclones, the largest encompassing circle in each clonal relationship represents the trunk in the phylogenetic tree, while smaller concentric circles illustrate subsequent waves of clonal expansion. We adjusted the circle areas based on the ratio of phylogenetically related cells to the total cell population within each circle.

In theory, if all cells in a biopsy exhibit a common lineage, the resultant circle would exceed the boundaries of the square due to its geometrical properties. Thus, in cases where any of the circles exceeded the boundary, we proportionally reduced the dimensions of all the inner circles to fit within the outer square's limits.

### Irradiation of melanocytes and keratinocytes in vitro (Related to Fig. S4)

Primary melanocytes (Fisher Scientific, C0025C) and keratinocytes (Fisher Scientific, C0015C) were commercially purchased. The cell lines were derived from the foreskin of newborn donors. These cells were cultured in their previously defined respective media under standard conditions at 37 °C in a humidified incubator with 5% $CO_2$. Once cultures were well established, 5000 cells of each type were seeded per well in 24-well plates ($n = 3$ wells per cell type per condition) and allowed to adhere and recover for 48 hours.

Next, cells were exposed to simulated solar UV radiation using the LS1000-4S-AM Solar Simulator (Solar Light LLC), which emits a full-spectrum light approximating natural sunlight. The biologically effective UV doses were quantified with the Digital Biologically Weighted Erythema UVB Sensor (Model PMA2101, Solar Light LLC), positioned 5 inches from the light source and oriented directly toward the lamp. This sensor, with a spectral response range of 290–400 nm permits accurate measurement of UVA, UVB, and combined UVA + UVB (replicating solar radiation, calibrated to the human erythema action spectrum).

Cells from each cell type were subjected to three experimental conditions: no irradiation (control), 5-minute UV exposure (-500 J/m²), and 10-minute UV exposure (-1000 J/m²), corresponding to approximately 0, 0.45, and 0.89 Minimal Erythema Dose (MED), respectively. Higher UV doses were found to be cytotoxic and resulted in complete cell death across all cell types. Cell counts were performed on Day 1 (pre-irradiation), 3, and 5 (post-irradiation) to assess cell viability and proliferation over time.

For downstream mutational profiling, melanocytes and keratinocytes from each condition (no irradiation, 5-minute, and 10-minute exposures) were isolated via single-cell sorting, clonally expanded, and processed for sequencing as described earlier. In total, 13 keratinocyte clones ($n = 5$ for 0 minutes, $n = 3$ for 5 minutes and $n = 5$ for 10 minutes of exposures) and 12 melanocyte clones ($n = 4$ per condition) were sequenced to evaluate UV-induced mutational burden.

### Dissection of neoplastic tissues for profiling squamous cell carcinomas in association with actinic keratoses

Twenty cutaneous squamous cell carcinomas with an adjacent actinic keratosis were retrieved from the archive of the Dermatopathology Service at UCSF. All skin excisional biopsies exhibited solar elastosis graded between 2+ and 3 + [69]. Biopsy sizes varied depending on the clinical requirements of the procedure. Breslow thickness ranged from 1.1 to 2.2 mm. Detailed case-specific clinicopathological features, collected with donor consent in accordance with UCSF IRB approvals, are provided in Supplementary Data S4. Based on morphological assessment by dermatopathologists, all squamous cell carcinoma tumor cells were determined to be well differentiated.

H&E scanned images were marked by a pathologist to identify the areas of squamous cell carcinoma, actinic keratosis, or non-neoplastic tissue (used as a genetic reference). Consecutive unstained sections (10 slides at 10 μm thickness) were dissected with a scalpel under a dissection scope to separate the histopathologically distinct tissues. Genomic DNA was isolated using the QIAamp DNA FFPE Tissue Kit (Qiagen, 56404). In four cases, the DNA yields were insufficient in one of the tissue areas, and these cases were discarded from future analyses. The remaining sixteen cases were retained for sequencing. This study was approved by the UCSF Human Research Protection Program, and all tissues were collected in accordance with the Institutional Review Board.

### Immunohistochemistry (IHC) staining

P53 immunohistochemistry (IHC) staining (clone DO-7 mouse, Roche, 1:400 dilution) was performed by the UCSF Dermatopathology Service following a clinically standardized protocol[70]. Phospho-MAPK was detected using the Phospho-p44/42 MAPK (Erk1/2) (Thr202/Tyr204) antibody (clone 4370, Cell Signaling, 1:100 dilution) by the UCSF Histology and Biomarkers Core on an automated Ventana BenchMark Ultra platform. The primary signal was visualized using the Discovery Purple detection kit. Images were captured using a Zeiss Axio Scanner

Z1 slide scanner equipped with a 20X Plan-Apochromat objective and processed with Zeiss Zen Lite v3.6 software.

## DNA sequencing and somatic alteration calling from neoplastic tissues

DNA sequencing and the initial steps of bioinformatic analyses were performed by the UCSF Clinical Cancer Genomics Laboratory (CCGL)[71]. CCGL is a CLIA-approved laboratory that performs gene-panel sequencing of tumors to help guide targeted treatments and assist in diagnosis. Specifically, 20-250 ng of genomic DNA was prepared for sequencing using the KAPA HyperPrep Kit with Library Amplification (Roche, KK8504). Target enrichment with a customized bait panel targeting 538 cancer-relevant genes (Supplementary Data S5) was performed using NimbleGen SeqCap EZ Developer library (Roche, Ref: 06471706001). Sequencing was performed on an Illumina HiSeq 2500 instrument. Alignment and grooming were performed with Burrows-Wheeler Aligner (BWA)[57], Genome Analysis Tool-Kit (GATK)[72], and Picard (https://broadinstitute.github.io/picard/). Copy number inference was performed with CNVkit[65,66].

We used two separate approaches to call somatic point mutations. First, we ran MuTect (v4.1.2.0) by comparing the tumor BAM files to reference BAM files from non-lesional tissue of the same patient. We filtered out all variants with fewer than 4 reads. In a typical tumor/normal sequencing study, MuTect is sufficient to call point mutations. However, in our study, the reference BAM often had small numbers of mutant reads. Mutant reads were common in normal-appearing tissue of our cohort because occult fields of keratinocytes clonally related to the neoplasm sometimes extended into the histologically normal skin, as we previously demonstrated to be a feature of keratinocyte cancers[43]. MuTect tends to reject variants with reads in the reference BAM, even under permissive parameters. MuTect is also not designed to call indels.

To supplement the MuTect calls, we also called variants against the reference genome using UnifiedGenotyper (v4.1.2.0) and FreeBayes (v1.3.1–19)[73]. These variant callers were incorporated to identify any point mutations missed by MuTect. We discarded variants from these lists, which were likely to be germline SNPs. We accomplished this goal by removing variants that resided in known SNP sites (from 1000 genomes) and/or had an allele frequency in the normal tissue of greater than 20%. We also discarded variants that were likely to be sequencing and/or alignment artifacts if they were observed in the blacklist of artifacts, previously defined by our cancer center's clinical cancer genomics laboratory from a list of recurring variants in panels of hundreds of normal tissues that had been sequenced through their services. The variants that were not filtered out in the above operations were considered somatic mutations and added to the list of somatic variants called by MuTect.

We called indels using Pindel. Candidate indels with greater than 4 supporting reads in the tumor but not the normal were manually inspected. The final list of somatic point mutations and indels is available in Supplementary Data S6.

## Inference of cancer genome fraction in neoplastic tissues (related to Supplementary Data S4)

Tumor genome fraction was inferred bioinformatically using multiple strategies. A short name for each strategy is listed in columns E through I of Supplementary Data S4 and described in more detail below.

Allelic Imbalance: Allelic imbalance over stretches of heterozygous SNPs is introduced when copy-number-neutral (CNN) loss-of-heterozygosity (LOH) or a deletion occurs in a tumor cell. In sequencing reads originating from tumor cells, the percentage of reads from either allele shifts to 100/0, but remains 50/50 from sequencing reads originating from stromal cells. For fully clonal LOH, the extent of allelic imbalance is therefore proportional to the tumor genome fraction (see Shain et al. NEJM, 2015[71] for the specific formula used).

Average Somatic MAF Autosomes: Somatic mutations can be stratified by their mutant allele frequencies (MAFs), which are dictated by the clonality and the zygosity of the mutation. Here, we used the median MAF of somatic mutations occupying portions of the genome without copy number alterations to infer tumor purity. This approach assumes those mutations are fully clonal and heterozygous. In some cases, there was a bimodal distribution of mutant allele frequencies with one population of mutations having notably low allele frequencies, likely stemming from subclones (also see the cross-contamination note below). When this occurred, we only considered, for this calculation, the population of mutations that we believed to be fully clonal. After calculating the median MAF of clonal mutations, we multiplied this value by 2 to arrive at the tumor genome fraction.

Driver Mutation: Some samples had few mutations, precluding the usage of the 'Median Somatic MAF Autosomes' method of inference described above, but every sample had at least one driver mutation. The mutant allele frequency of the driver mutation was used to estimate tumor purity under the assumption that the mutation was heterozygous and fully clonal – before making these assumptions, we checked for loss-of-heterozygosity or a copy number alteration affecting the locus of the driver mutation. We multiplied the MAF of driver mutations by 2 to arrive at the tumor cellularity estimate.

Median Somatic MAF XY: In a male sample, a somatic mutation on the X or Y chromosome will have a mutant allele frequency 100% from sequencing reads derived from the tumor cells. Sequencing reads from stromal cells will not contribute any mutant reads. The observed mutant allele frequency of the mutation can therefore be used to infer the relative proportions of tumor and stromal cells. This approach assumes these mutations are fully clonal and do not reside in chromosomes with copy number alterations.

## Cross-contamination estimates (Related to Fig. 3b, S9, and Supplementary Data S4)

In column J of Supplementary Data S4, we include a column with a cross-contamination note. Some of the squamous cell carcinomas show signs of contamination with actinic keratosis cells or vice-versa. Typically, contamination was evident when mutations of the squamous cell carcinoma had trace sequencing reads in the actinic keratosis (or vice-versa). The level of trace sequencing reads was used to infer degrees of cross-contamination by doubling their allele frequencies (which assumes that the median contaminant mutation is fully clonal and heterozygous).

## Allelic imbalance calculation from tumor sequencing data (related to Fig. 3e)

To measure allelic imbalance in the sequencing data from squamous cell carcinomas and actinic keratoses, we first identified a set of heterozygous SNPs for each patient as described above. Once a set of high-confidence SNPs was derived, we counted the ref and alt reads of each SNP in the sequencing data of the squamous cell carcinoma and actinic keratosis. Next, we calculated the variant allele fraction of the major allele (i.e., the more abundant allele in the sequencing data) and subtracted 0.5 (the expected fraction if the allele were sampled equally in the sequencing data). We plotted allelic imbalance values across the genome for each sample to identify contiguous regions with imbalance relative to the background values.

## Annotation of pathogenic mutations in squamous cell carcinomas in association with actinic keratosis

See the section entitled, Annotation of pathogenic mutations in individual cells, above, for a description of this process. The full list of mutations is available in Supplementary Data S6 for interpretation, and our annotation of pathogenic mutations can be found in column W.

## Phylogenetic tree construction for neoplastic tissues (Related to Figs. 3f, 4, S10 and S11)

We constructed phylogenetic trees for the 5 squamous cell carcinomas that evolved from the neighboring actinic keratoses (shown in Figs. 3f and 4a). Mutations were categorized as shared or private between the dominant clones in each area and were respectively placed on the trunk or branch of each tree. Our determinations of trunk vs branch mutations are shown in the color-coded scatterplots of Fig. 3b and S9, and below we describe how these calls were made.

Classifying mutations as shared or private was challenging due to low tumor cell content and the presence of cross-contamination between different neoplastic areas. To take these factors into account, a mutation was considered present in the dominant clone of the AK or the SCC when it was more than 50% clonal. The clonality of a point mutation was estimated by its allele frequency relative to tumor purity and after accounting for cross-contamination and the copy number/zygosity of the mutation. Despite establishing this cutoff, there were instances in which mutations from unrelated clones of keratinocytes appeared to be incorporated into the phylogenetic trees, requiring further refinement on a sample-by-sample basis. To further refine the mutations assigned to each clone, we generated histograms of mutant allele frequencies for the actinic keratosis and squamous cell carcinoma areas. Histograms tended to be multimodal, and we assumed each peak corresponded to the median mutant allele frequency of mutations in a given clone. We identified breaks between peaks in the histograms to remove clusters of low allele frequency mutations that likely stemmed from unrelated clones of keratinocytes.

Due to the complex clonal structure of the skin samples, we elected not to call branch mutations in actinic keratoses. The tumor purities of actinic keratoses were low, and this made it difficult to distinguish clusters of mutations in the actinic keratosis from those originating from unrelated clones in the area.

We also constructed phylogenetic trees for the cutaneous squamous cell carcinomas and adjacent skin samples from Kim and colleagues[18]. Since their gene panel was relatively small, fewer mutations were available to calculate tumor cell content and clonality of mutations. Instead, we generated scatterplots of mutation allele frequencies in for different progression stages (see Fig. S10 and S11) and manually categorized mutations as shared or private based on how they clustered in these plots. Our phylogenetic trees are shown side-by-side with the original trees from Kim and colleagues.

## Comparing the genomic landscape of keratinocytes inferred from a single-cell and bulk sequencing workflow (Related to Fig. S6)

To compare the mutational landscape inferred through our single-cell workflow with that from standard bulk sequencing, we performed both methods on patient-matched biopsies. These biopsies were collected from a deceased 74-year-old male donor (D56), identified as of European ancestry, and were part of this study via the UCSF Willed Body program. We collected 5 mm and 3.5 mm punch biopsies from both the face and the shoulder. The 5 mm biopsies were used to establish a bulk primary culture, followed by the subsequent single-cell sequencing workflow. A total of 15 keratinocytes from the face and 13 keratinocytes from the shoulder were sequenced following the procedures described earlier (Figs. 1, 2, S3–S7). The 3.5 mm biopsies were used for bulk-cell sequencing of epidermis. Specifically, for the 3.5 mm biopsies, the epidermis and dermis were separated with dispase treatment (10 mg/ml for 16 hours). Bulk DNA was isolated from the epidermis using the prepIT.L2P kit (DNA Genotek, PT-L2P-5). This bulk DNA was directly used for sequencing library preparation. Library preparation and hybridization were performed identically to the corresponding single-cell samples, utilizing KAPA HyperExome (Roche, 09062556001) probes for exome enrichment. Sequencing (100 bp, paired-end) was performed on the NovaSeq 6000 platform. Somatic

mutations were called using MuTect and Pindel, as described above. The mutation burden per cell was estimated from bulk-cell sequencing data following a method similar to that described by Martincorena et al.[10]. In their approach, mutations in normal skin are parsimoniously assumed to be heterozygous. We also made this assumption for autosomal mutations, but we made a slight modification to account for the non-diploid nature of sex chromosomes in male samples.

The clone size ($\rho$) was estimated as:

$$\rho = 2^{*}VAF\text{(autosomal region)and} \qquad (1)$$

$$\rho = VAF\text{(X and Y chromosome)} \qquad (2)$$

where VAF is the variant allelic frequency of mutations in those regions.

The mutation burden per cell ($\beta$) was calculated as:

$$\beta = \Sigma(\rho)/L_{Mb} \qquad (3)$$

Here, $\Sigma(\rho) \approx \Sigma(VAF_{X/Y}) + 2^{*}\Sigma(VAF_{autosomal})$, where $VAF_{X/Y}$ represents VAF of mutations in sex chromosomes and $VAF_{autosomal}$ represents VAF of mutations in autosomal regions. $L_{Mb}$ is the footprint (genomic region with good coverage). The $L_{Mb}$ for KAPA HyperExome has been thoroughly validated as 43.2 Mb.

Mutation signature analysis was performed on an aggregated list of the mutations from the 15 keratinocytes sequenced from the face, the 13 keratinocytes sequenced from the shoulder, and mutations inferred in adjoining face and shoulder biopsies. The same single-cell pipeline described earlier was applied for this analysis.

## Spatial transcriptomics (Related to Fig. 5, S12, S13, and S14)

Spatial transcriptomics was performed on five squamous cell carcinomas in association with actinic keratoses. We chose a subset of the cases that underwent DNA-sequencing, as described above, so that matching mutational and spatial transcriptomic information would be available. In four of the five cases, the squamous cell carcinoma was genetically related to the neighboring actinic keratosis (BB05, BB12, BB13, and BB16), and in the remaining case, the squamous cell carcinoma was not genetically related to the neighboring actinic keratosis (BB09). All cases were profiled on a version of the 10X FFPE Visium platform.

One case (BB13) was profiled on a relatively older version of Visium (v1.0). For this case, we cut additional sections of the tissue from its original block and placed them within the fiducial frame on the slide. The slide was prepared according to the manufacturer's protocols at UCSF. The remaining cases (BB05, BB09, BB12, and BB16) were profiled with a relatively newer version of Visium (v2.0) that is compatible with a CytAssist machine (10X Genomics). For these cases, we took an existing H&E slide, removed the coverslip, and situated the tissue within the designated capture area on the CytAssist machine, where probe hybridization occurred. Hybridization and preparation for sequencing were performed according to the manufacturer's protocols by an outside company, Abiosciences.

Paired-end sequencing was performed by the Center for Advanced Technology at UCSF on an Illumina instrument (NovaSeq 6000). Read 1 (the barcode read) was sequenced with a read length of 28 bp, and read 2 (the probe read) was sequenced with a read length of 90 bp. Sequencing data was processed with the SpaceRanger pipeline (v1.3.0 and v2.0.1) to generate a cloupe file, which was visualized in the Loupe browser (version 7, 10X Genomics). The SpaceRanger workflow can run samples one-by-one or in aggregate mode. The data from the four samples (BB05, BB09, BB12, and BB16) that were run on the relatively newer version of Visium were merged into an aggregate run. The sequencing data from the remaining sample (BB13), which was run

on a relatively older version of Visium, could not be merged with the others due to differences in the chemistry of the platforms.

For the copy number analyses described in Fig. S12, we used STmut to infer copy number from individual spots[43]. Briefly, we ran STmut in grouping mode, which can combine contiguous spots from the same gene expression cluster that have fewer than 1000 genes detected. After combining spots with low coverage, the groups of spots are treated as a single spot. In practice, most spots had more than 1000 genes detected and were not affected by this parameter, but this feature improved the signal-to-noise in a subset of spots with low sequencing coverage. STmut can also accept lists of known copy number alterations and generate q-values for a given spot, which reflects the likelihood that it matches a known copy number profile. We called arm-level gains and losses from DNA-sequencing data and input these calls into STmut. The histograms and Q-Q plot in Fig. S12 show the spots with high CNVscores (i.e., high similarity to the known copy number profile), which we considered to be spots overlying squamous cell carcinoma.

For the gene expression analyses described in Fig. S13b and d, we used the graph-based clusters generated by the SpaceRanger software. The identities of the clusters were manually annotated after examining the histology of the underlying cells and their gene expression patterns. Differentially expressed genes across the spots in these clusters were exported from the Loupe browser. For the gene expression cluster in Fig. S13c, we manually annotated areas of squamous cell carcinoma or actinic keratosis and performed differential gene expression analyses. To annotate squamous cell carcinoma versus actinic keratosis, we considered copy number data and histology to make the final calls.

### Reporting summary

Further information on research design is available in the Nature Portfolio Reporting Summary linked to this article.

## Data availability

This study is part of the Human Tumor Atlas Network (HTAN), which is funded by the National Cancer Institute (U01 CA294536). The goal of HTAN is to catalog molecular transitions during the evolution of cancer. Raw and intermediate data are immediately available, as described below. These data will also be accessible through the HTAN data portal after the next data release (currently anticipated for Spring of 2025). The DNA and RNA sequencing data of individual skin cells is available in dbGaP under accession codes phs001979.v1.p1 [https://www.ncbi.nlm.nih.gov/projects/gap/cgi-bin/study.cgi?study_id=phs001979.v1.p1] and phs003683.v2.p1 [https://www.ncbi.nlm.nih.gov/projects/gap/cgi-bin/study.cgi?study_id=phs003683.v2.p1]. The DNA sequencing data and spatial transcriptomic data from the cutaneous squamous cell carcinomas in association with actinic keratoses are available in dbGaP under accession code phs003282.v2.p1 [https://www.ncbi.nlm.nih.gov/projects/gap/cgi-bin/study.cgi?study_id=phs003282.v2.p1]. These accession numbers provide access to the raw sequencing FASTQ files. Access to these datasets is restricted because participant consent permits data use only for biomedical research and does not allow unrestricted public release of individual-level genomic information. Investigators can request access through the dbGaP Data Access Committee via the dbGaP portal, and approved users receive data under institutional approvals and data use agreements consistent with the original consent. Requests are typically reviewed within 4–8 weeks, and data remain available for the duration of the repository's retention policy. Intermediate levels of analysis are also available. Somatic mutation calls for individual cells are available in Supplementary Data S3 and were deposited in cBioPortal [https://www.cbioportal.org/study/clinicalData?id=normal_skin_keratinocytes_2024]. A summary of genetic alterations in each keratinocyte, as well as copy number data

from each cell is available on figshare: https://figshare.com/projects/Genetic_evolution_of_keratinocytes_to_cutaneous_squamous_cell_carcinoma/199837. Publicly available mutation data, covering the progression of squamous cell carcinoma from potential precursor lesions, was retrieved from Supplementary Table S3 of Kim et. al., *JID*, 2022[18]. Publicly available mutation data, covering the somatic point mutations in epidermal biopsies, was retrieved from Supplementary Table 2 (NIHMS63718-supplement-2.xlsx) of Martincorena et. al. *Science*, 2015[10]. All analyses on publicly available data were performed with appropriate citation of the original source. Source data are provided with this paper as the source data file. Source data are provided with this paper.

## Code availability

All data analyses were conducted using publicly available software packages. The pipeline and tools for somatic mutation calling is available at [https://github.com/ShainLab/Single_Cell_Somatic_Mutation_Caller], a coverage analysis tool for counting the number of bases in the bam file with a specified coverage at [https://github.com/ShainLab/Footprints_v.0.1], script for identifying heterozygous Single Nucleotide Polymorphisms and haplotype phasing at [https://github.com/ShainLab/HaploPrep], the phylogenetic tree construction script at [https://github.com/ShainLab/Phylogenetic_tree], and the spatial transcriptomics package Stmut at [https://github.com/ShainLab/STmut]. All codes are publicly available under the MIT License.

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

## Acknowledgements
We thank all the donors from whom we obtained specimens and sequencing data. The study was supported by grants from: NIH NCI Human Tumor Atlas (HTAN) network (U01 CA294536 - AHS, IY, BCB), NIH NCI (R01 CA265786, AHS), NIH NIAMS (AR080626, AHS), Department of Defense Melanoma Research Program (ME210014, AHS), Melanoma Research Alliance Team Science Award (AHS), Melanoma Research Alliance Dermatology Fellows Award (BT), the LEO Foundation Region Americas Award (AHS), a private donation by Tracy and Guy Jaquier (AHS), and the UCSF Department of Dermatology.

## Author contributions
A.H.S. was responsible for conceptualization. Data curation was performed by A.H.S., B.T., D.D., L.C., and S.T.A. Formal analysis and methodology were carried out by A.H.S., B.T., DD, and L.C. Investigation was conducted by A.H.S., B.T., and D.D. L.C., N.C.P., H.S., A.X., A.K.B., D.B.C., C.G., A.M., R.J.C., J.C., D.S., P.G., S.T.A., B.C.B., M.W., and I.Y. Visualization was performed by A.H.S., B.T., D.D., and M.W. Funding acquisition was secured by A.H.S., B.T., B.C.B., R.J.C., J.C., and D.S. Project administration was managed by A.H.S., and resources were provided by A.H.S., B.C.B., P.G., S.T.A., and J.C. Supervision was provided by AHS. The original draft was written by A.H.S., B.T., and D.D., and all authors contributed to the review and editing of the manuscript.

## Competing interests
AHS has an industry-sponsored research agreement with Kenvue that does not pertain to the work described here. The remaining authors declare no competing interests.
