## [Peer Review File · Nature Communications]

Genetic evolution of keratinocytes to cutaneous squamous cell carcinoma

Corresponding Author: Dr Alan Shain

This file contains all reviewer reports in order by version, followed by all author rebuttals in order by version. Parts of this Peer Review File have been redacted as indicated to remove third-party material.

Version 0:

Reviewer comments:

Reviewer #2

(Remarks to the Author)

I appreciate the opportunity to review this excellent article, which is worthy of publication in this journal with minor revisions. The study's data analysis and interpretation are sound, there is enough detail provided in the methods for the work to be reproduced, and the conclusions are well-supported by the evidence provided. To better understand the molecular transitions during skin carcinogenesis, the authors profiled the mutational and transcriptional landscapes of individual keratinocytes from clinically normal human skin. The authors have made significant progress in understanding the mutational events that occur from apparently normal skin to actinic keratoses and squamous cell carcinomas, challenging some existing assumptions in the field. For instance, it was observed that most keratinocytes have low mutation burdens despite extensive sun exposure. However, keratinocytes with mutations in TP53 or NOTCH1 exhibited substantially higher mutation burdens. Also, the study found that squamous cell carcinomas (SCCs) are often genetically unrelated to neighbouring precursor lesions such as actinic keratoses (AK). They also performed DNA sequencing and spatial transcriptomics of actinic keratoses adjacent to cutaneous squamous cell carcinomas. The authors created new data and reanalyzed existing data to compare their findings with those of previously published studies. This article significantly advances the field of skin cancer research by providing new insights into the genetic evolution of keratinocytes and the complex interactions within the tumour microenvironment. While the study provides substantial evidence for its claims, further research could strengthen these conclusions. Additional studies with larger sample sizes, diverse populations and different histological types of cSCC could help confirm the generalizability of the findings.

There are multiple exciting findings, some of them new and other findings that prove what was previously known:

-In single-cell clonal expansions from normal human skin: They compared the median mutation burden of keratinocytes, melanocytes and fibroblasts obtained from single-cell expansions (the lineage of the cell type was confirmed). An unexpectedly low mutation burden of keratinocytes compared to melanocytes and fibroblasts was detected. Keratinocytes with missense mutations in TP53, had the highest mutation burdens. These findings suggest that wild-type keratinocytes are remarkably well-adapted to repair DNA damage or undergo cell death when levels of DNA damage are beyond repair, compared to melanocytes, as previously published. Autosomal copy number alterations were infrequent. Cells with defective p53 are thus likely to accumulate DNA damage at a higher rate. NOTCH1 mutations induce a stem/progenitor cell state in epithelial cells, which may also prevent cells from undergoing apoptosis upon excessive DNA damage. Loss-of-function mutations in TP53 and NOTCH1 are known to provide a fitness advantage to epithelial cells, but findings from this article suggest their contribution to tumour progression may primarily stem from the mutator phenotypes that they induce. The authors suggest that the lower mutation burden of keratinocytes without these mutations compared to epidermal melanocytes and dermal fibroblasts may result from a lower threshold of keratinocytes to undergo apoptosis. It is an interesting hypothesis that still needs to be proven. Mutational signature analyses revealed differences in the types of mutations between keratinocytes, melanocytes, and fibroblasts. Interestingly, fibroblasts had significant UV radiation signature mutations.

-They tried to minimize the bias inherent to single-cell genotyping, comparing the mutational landscapes of individual keratinocytes to bulk cell sequencing of epidermis from previous and new and own data. While the types of mutations in this study were similar to bulk-cell measurements, the numbers of mutations per cell were somewhat lower. One interesting and possible explanation is that keratinocytes with low mutation burdens are more likely to expand clonally. Some keratinocytes shared a portion of their somatic mutations with other keratinocytes from the same biopsy, indicating that they are clonally related. Keratinocyte clones were more prevalent than melanocyte clones and less likely to harbour pathogenic mutations. Interestingly, clones of keratinocytes with pathogenic mutations were not bigger than clones of keratinocytes without

pathogenic mutations.

-The authors studied the genetic alterations driving the transition from actinic keratosis immediately adjacent to squamous cell carcinoma from DNA obtained from scalped dissected tissue (FFPE) from 16 patients, using non-neoplastic tissue as reference. They analyze 538 cancer-relevant genes using deep sequencing. In 6 of the 16 cases, the actinic keratosis and adjacent squamous cell carcinoma did not share somatic alterations. In 4 out of 16 cases, there were no mutations exclusive to squamous cell carcinoma, and in 5 out of 16 cases, squamous cell carcinomas evolved from the neighbouring actinic keratosis. Then, it is not safe to assume that keratinocyte lesions nearby are necessarily clonally related. The authors annotated mutations in genes known to drive keratinocyte cancers (it should be noted that there are other mutations in common, but they are not shown in the article and can be found in the supplementary tables). Based on the distribution of shared and unshared somatic alterations in the dominant clones, they inferred the order in which mutations occurred. -They complemented the cohort, reanalyzing publicly available data and finding similar results. Some squamous cell carcinomas do not share mutations with neighbouring tissue, and some evolved from neighbouring precursor lesions (actinic keratosis, squamous cell carcinoma in situ or sun-exposed skin). Mutations in the p53, Notch, TERT, and Rb pathways tended to occur early (some happen even before neoplasms are present), contributing to forming actinic keratoses. Mutations affecting the SWI/SNF chromatin remodelling complex or activating the MAPK/PI3K pathways tended to occur later, driving the transition to squamous cell carcinoma.

-The amount of tissue that can be obtained from a skin tumour is very small compared to other tumours, e.g. breast, lung, gastric cancer, etc. They cleverly used spatial transcriptomic analysis in order to take maximum advantage of the available tissue. The spatial transcriptomic analysis (10X Visium) on five of the squamous cell carcinomas adjacent to actinic keratoses (clonally related) was performed. It helped define the localization of tumour cells, revealing a complex spatial architecture of clones. They could identify spots with copy number alterations similar to those observed in bulk-cell DNA-sequencing data. Interestingly, the spatial data revealed satellite colonies of cells physically distant from the primary tumour. It may explain higher recurrent rates in cSCC compared with basal cell carcinoma. Gene expression programs spanned a range of differentiation states within actinic keratoses and squamous cell carcinomas. Spots with stem-like signatures were exclusive to the invasive front of squamous cell carcinoma. They express mesenchymal genes, which are typically observed during brief periods of epithelial development or after injury. However, hierarchies of differentiation are maintained, even in fully evolved tumours. Concerning this point, it is necessary, as I request below, to describe some minimal histopathological characteristics of the cSCCs incorporated in the study.

-Finally, they studied the immune infiltrates to extend along the borders of actinic keratosis and squamous cell carcinoma. They found higher immune checkpoint ligands and protein expression in the lymphocytes at the invasive front of squamous cell carcinomas.

There are some aspects that need to be clarified:

-The authors selected multiple donors to measure somatic mutations in single-cell expansions. For this, they collected biopsies from body sites that experience different degrees of habitual sun exposure, including the buttocks, trunk, and head/neck area. Apparently, the level of sun exposure was considered based on age and anatomic location alone. Sun exposure will depend on multiple factors: occupation, sporting activities, Fitzpatrick phototype, ethnic origin, etc. It would be interesting to know if there is a clinical history of the degree of sun exposure, clinical signs and/or if there are histologic indirect markers of the degree of sun exposure, such as the degree of solar elastosis. Ethnicity could be deduced from genomic data. Some of the data should be interpreted based on this new information on some objective measure of sun exposure. If not, this aspect should be considered a work limitation, and the authors should explain how much it could affect the results.

On page 2, the last paragraph, it says, "individual keratinocytes from physiologically normal human skin." I think it could say, "individual keratinocytes from clinically normal human skin." One could also say histologically normal if histology had been analyzed.

-On page 4, reference 29 is from a review article and not the one with research data

-In methods, the available SCC clinicopathological features should be described (i.e., biopsy type, clinical size, thickness, degree of differentiation, etc.). It seems that most of the samples were taken by tissue saucerization and are well-differentiated cSCC. With this new information, evaluate whether modifying or clarifying any of the statements is necessary. For example, the one related to the cSCC differentiation spectrum in the spatial transcriptomic data analysis.

-On page 5, the authors suggest that in 4 cases, there were no mutations exclusive to the squamous cell, implying a single population of clonally related cells spanning both dissected tissue areas, with no identifiable mutations accounting for the progression to invasive carcinoma. For these cases, the actinic keratosis histology may represent an extension of the squamous cell carcinoma rather than a distinct precursor lesion. I would like you to clarify this idea better. Are there other possibilities? (ie. Epigenetics, methodologic bias, non-detected mutations on cSCC, etc).

-On page 5, the authors show that higher phospho-MAPK signalling was observed in the squamous cell carcinoma (Fig. 3H), consistent with the CBL mutation. I think it is a very indirect way to show the effect of CBL mutation. MAPK ERK could be activated for many other different reasons. Please comment on this point and defend your position

- Regarding figure 6. Why does Hippo appear in red? (FAT1 is considered a tumour suppressor gene, and Hippo ON controls YAP/TAZ, inducing sequestering in the cytoplasm and degradation by the proteasome)

Reviewer #3

(Remarks to the Author)

This manuscript presents a comprehensive multi-omic analysis of the genetic evolution of cutaneous squamous cell carcinoma (cSCC). The authors utilize single-cell and spatial transcriptomic techniques to track the progression from normal keratinocytes to actinic keratosis and finally to cSCC. This approach provides novel insights into the clonal architecture of cSCC development and identifies key mutational events and gene expression changes during this process. The findings

challenge the traditional view of cSCC development, revealing unexpected complexities in clonal relationships, the role of specific mutations (TP53 and NOTCH 1), and the spatial heterogeneity of immune responses within tumors. The identification of frequent "collisions" of unrelated neoplasms is particularly noteworthy and is in accordance with the concept of field cancerization. The manuscript is well-written and the results are presented clearly, with appropriate use of figures and supplementary materials.

Specific comments:

While the number of samples analyzed is substantial, a larger sample size, particularly for the spatial transcriptomics, would strengthen the conclusions. The relatively low number of cases with a clear precursor-lesion relationship limits the power of those specific analyses. This should be commented.

Identification of loss-of-function mutations of TP53, NOTCH 1, NOTCH2, and CDKN2A and gain-of-function mutation of the TERT promoter in cSCC are in accordance with previous observations and not novel. However loss-of-function mutations in ARID2 and CBL in cSCC are interesting and should be discussed further e.g. in terms of their functional role in cSCC progression. Also the role of mutations resulting in the activation of MAPK pathway, e.g. in HRAS should be discussed.

The authors acknowledge the limitations of clonal expansion in introducing bias and the challenges of accurately calling somatic mutations in individual cells. Discussion of potential biases inherent in the spatial transcriptomics data is also warranted.

The authors should further discuss the clinical implications of the results for diagnosis, prognosis, and treatment strategies.

Reviewer #4

(Remarks to the Author)

The manuscript has generated genomic profiles of multiple skin cell types, including keratinocytes, melanocytes, and fibroblasts across normal skin, precancerous lesions, and squamous cell carcinoma samples. The authors have shown that keratinocytes can cope with UV well, with fewer mutations accumulating than melanocytes. They have also shown the genomic path for the evolution of squamous cell carcinoma, especially by comparing the precancerous lesion and cancer area taken from the same patients. Finally, they match their finding with spatial transcriptomics analysis on the same tissue, which shows the enrichment of activated/exhausted immune cells in SCC area.

One of the most interesting and novel findings from this study is the low mutational burden of keratinocytes. While the authors have suggested a potential mechanism for this, there's no experimental validation, which limits the impact of the current finding. As the authors have set up the culture conditions for these cell types, the hypothesis can be tested by UV exposure and clonal evolution analysis in vitro.

In addition, the authors have tried to combine the spatial transcriptomics (10X Visium) with the clonal evolution. However, the current scope of spatial analysis is of low resolution, with simple sketches with gene set signature scores. Except for the fact that there are more activated/exhausted immune cells enriched around the SCC area compared to AK area, which is already well known and expected, the spatial transcriptomics do not offer any advancement in our understanding of tumor microenvironment evolution.

Version 1:

Reviewer comments:

Reviewer #2

(Remarks to the Author)

After reviewing the new version of the manuscript, I can conclude that the authors have incorporated the suggestions I requested, performed new analyses of their data, and conducted new experiments where necessary. They have also acknowledged the limitations of their study and adjusted their conclusions to reflect the new evidence. In my opinion, the article should be accepted for publication in your journal.

Reviewer #3

(Remarks to the Author)

The authors have revised the manuscript in response to criticism raised and it has greatly improved. I have no further comments.

Reviewer #4

(Remarks to the Author)

The authors have addressed my comments with a well-designed experiment, which adds a mechanical aspect to the paper. I recommend publication of this paper as is.

Thank you to the editor and reviewers for your time and helpful feedback, which has improved the current version of our manuscript. The most significant change to this revision is the addition of figure S4, where we irradiate primary melanocytes and keratinocytes *in vitro*. The results (discussed in detail below) are a starting point towards learning how UV radiation induces cancer in different lineages of cells but fully delineating these mechanisms would require standalone paper. To illustrate our point, the relatively simple irradiation experiment shown in figure S4 involved exome/transcriptome sequencing of 25 subclones of cells. In addition to the experimental work in figure S4, we report new data and made editorial changes in response to reviewer-specific critiques, as detailed in the point by point responses below.

Reviewer #2 (Remarks to the Author):

I appreciate the opportunity to review this excellent article, which is worthy of publication in this journal with minor revisions. The study's data analysis and interpretation are sound, there is enough detail provided in the methods for the work to be reproduced, and the conclusions are well-supported by the evidence provided. To better understand the molecular transitions during skin carcinogenesis, the authors profiled the mutational and transcriptional landscapes of individual keratinocytes from clinically normal human skin. The authors have made significant progress in understanding the mutational events that occur from apparently normal skin to actinic keratoses and squamous cell carcinomas, challenging some existing assumptions in the field. For instance, it was observed that most keratinocytes have low mutation burdens despite extensive sun exposure. However, keratinocytes with mutations in TP53 or NOTCH1 exhibited substantially higher mutation burdens. Also, the study found that squamous cell carcinomas (SCCs) are often genetically unrelated to neighbouring precursor lesions such as actinic keratoses (AK). They also performed DNA sequencing and spatial transcriptomics of actinic keratoses adjacent to cutaneous squamous cell carcinomas. The authors created new data and reanalyzed existing data to compare their findings with those of previously published studies. This article significantly advances the field of skin cancer research by providing new insights into the genetic evolution of keratinocytes and the complex interactions within the tumour microenvironment. While the study provides substantial evidence for its claims, further research could strengthen these conclusions. Additional studies with larger sample sizes, diverse populations and different histological types of cSCC could help confirm the generalizability of the findings.

Thank you for the kind remarks and accurate summary of our manuscript.

We agree that additional studies with larger sample sizes would extend the generalizability of our findings, and we are actively pursuing these experiments on a much larger scale. However, this next phase will likely take several years, which goes beyond the scope of the current manuscript. Nonetheless, we believe the conclusions drawn from the present study are, as the reviewer notes, "worthy of publication" and "well-supported by the evidence provided".

There are multiple exciting findings, some of them new and other findings that prove what was previously known:

-In single-cell clonal expansions from normal human skin: They compared the median mutation burden of keratinocytes, melanocytes and fibroblasts obtained from single-cell expansions (the lineage of the cell type was confirmed). An unexpectedly low mutation burden of keratinocytes compared to melanocytes and fibroblasts was detected. Keratinocytes with missense mutations in TP53, had the highest mutation burdens. These findings suggest that wild-type keratinocytes are remarkably well-adapted to repair DNA damage or undergo cell death when levels of DNA damage are beyond repair, compared to melanocytes, as previously published. Autosomal copy number alterations were infrequent. Cells with defective p53 are thus likely to accumulate DNA damage at a higher rate. NOTCH1 mutations induce a stem/progenitor cell state in epithelial cells, which may also prevent cells from undergoing apoptosis upon excessive DNA damage. Loss-of-function mutations in TP53 and NOTCH1 are known to provide a fitness advantage to epithelial cells, but findings from this article suggest their contribution to tumour progression may primarily stem from the mutator phenotypes that they induce. The authors suggest that the lower mutation burden of keratinocytes without these mutations compared to epidermal melanocytes and dermal fibroblasts may result from a lower threshold of keratinocytes to undergo apoptosis. It is an interesting hypothesis that still needs to be proven.

Thank you again for the fantastic summary of our results. In response to your comment immediately above, “It is an interesting hypothesis that still needs to be proven”, we have added citations from other groups showing that epidermal keratinocytes are more likely than melanocytes to undergo apoptosis after sunburn (PMID: 12535197, 12856704, 3317295 and 10219070).

To complement these citations, and in response to reviewer 4, we also have added new experimental data to the revised manuscript. In figure S4, we performed in vitro irradiation of melanocytes and keratinocytes with the exact same doses of simulated solar UV radiation, roughly equivalent to one minimal erythral dose (MED) – the amount of UV radiation needed to induce a sunburn. These results show that the keratinocytes are more sensitive to UV radiation, however, the surviving keratinocytes accumulated more mutations than melanocytes (line 135-148, 666-691).

The in vitro irradiation results are fascinating but somewhat unexpected. How do keratinocytes in the skin maintain low mutation burdens while keratinocytes in 2-dimensional culture accumulate mutations efficiently? The 2-dimensional irradiation studies are informative but do not fully model mutation accumulation in a physiologically relevant microenvironment or on an appropriate time scale (ideally, decades of cumulative exposure rather than a single acute exposure).

It is possible keratinocytes with low mutation burdens recently departed from hair follicles. We show this to occur in melanocytes with abnormally low mutation burdens in heavily sun damaged skin (<https://www.biorxiv.org/content/10.1101/2025.02.07.637114v1>). It is also possible that the supranuclear cap of melanin is protective of keratinocytes in situ, but this is not present in 2-dimensional mono-culture of keratinocytes. These are interesting questions but will require additional studies, which go far beyond the scope of this manuscript. As such, we have tempered much of our speculation on how keratinocytes maintain low mutation burdens in the current version of the manuscript.

Mutational signature analyses revealed differences in the types of mutations between keratinocytes, melanocytes, and fibroblasts. Interestingly, fibroblasts had significant UV radiation signature mutations.

-They tried to minimize the bias inherent to single-cell genotyping, comparing the mutational landscapes of individual keratinocytes to bulk cell sequencing of epidermis from previous and new and own data. While the types of mutations in this study were similar to bulk-cell measurements, the numbers of mutations per cell were somewhat lower. One interesting and possible explanation is that keratinocytes with low mutation burdens are more likely to expand clonally. Some keratinocytes shared a portion of their somatic mutations with other keratinocytes from the same biopsy, indicating that they are clonally related. Keratinocyte clones were more prevalent than melanocyte clones and less likely to harbour pathogenic mutations. Interestingly, clones of keratinocytes with pathogenic mutations were not bigger than clones of keratinocytes without pathogenic mutations.

-The authors studied the genetic alterations driving the transition from actinic keratosis immediately adjacent to squamous cell carcinoma from DNA obtained from scalped dissected tissue (FFPE) from 16 patients, using non-neoplastic tissue as reference. They analyze 538 cancer-relevant genes using deep sequencing. In 6 of the 16 cases, the actinic keratosis and adjacent squamous cell carcinoma did not share somatic alterations. In 4 out of 16 cases, there were no mutations exclusive to squamous cell carcinoma, and in 5 out of 16 cases, squamous cell carcinomas evolved from the neighbouring actinic keratosis. Then, it is not safe to assume that keratinocyte lesions nearby are necessarily clonally related. The authors annotated mutations in genes known to drive keratinocyte cancers (it should be noted that there are other mutations in common, but they are not shown in the article and can be found in the supplementary tables). Based on the distribution of shared and unshared somatic alterations in the dominant clones, they inferred the order in which mutations occurred.

-They complemented the cohort, reanalyzing publicly available data and finding similar results. Some squamous cell carcinomas do not share mutations with neighbouring tissue, and some evolved from neighbouring precursor lesions (actinic keratosis, squamous cell carcinoma in situ or sun-exposed skin). Mutations in the p53, Notch, TERT, and Rb pathways tended to occur early (some happen even before neoplasms are present), contributing to forming actinic keratoses. Mutations affecting the SWI/SNF chromatin

remodelling complex or activating the MAPK/PI3K pathways tended to occur later, driving the transition to squamous cell carcinoma.

-The amount of tissue that can be obtained from a skin tumour is very small compared to other tumours, e.g. breast, lung, gastric cancer, etc. They cleverly used spatial transcriptomic analysis in order to take maximum advantage of the available tissue. The spatial transcriptomic analysis (10X Visium) on five of the squamous cell carcinomas adjacent to actinic keratoses (clonally related) was performed. It helped define the localization of tumour cells, revealing a complex spatial architecture of clones. They could identify spots with copy number alterations similar to those observed in bulk-cell DNA-sequencing data. Interestingly, the spatial data revealed satellite colonies of cells physically distant from the primary tumour. It may explain higher recurrent rates in cSCC compared with basal cell carcinoma. Gene expression programs spanned a range of differentiation states within actinic keratoses and squamous cell carcinomas. Spots with stem-like signatures were exclusive to the invasive front of squamous cell carcinoma. They express mesenchymal genes, which are typically observed during brief periods of epithelial development or after injury. However, hierarchies of differentiation are maintained, even in fully evolved tumours. Concerning this point, it is necessary, as I request below, to describe some minimal histopathological characteristics of the cSCCs incorporated in the study.

All of the summary above is accurate, and we agree with the immediate point above, “to describe some minimal histopathological characteristics of the cSCCs incorporated in the study”. We have added this data to table S4. (line 696-700)

-Finally, they studied the immune infiltrates to extend along the borders of actinic keratosis and squamous cell carcinoma. They found higher immune checkpoint ligands and protein expression in the lymphocytes at the invasive front of squamous cell carcinomas.

There are some aspects that need to be clarified:

-The authors selected multiple donors to measure somatic mutations in single-cell expansions. For this, they collected biopsies from body sites that experience different degrees of habitual sun exposure, including the buttocks, trunk, and head/neck area. Apparently, the level of sun exposure was considered based on age and anatomic location alone. Sun exposure will depend on multiple factors: occupation, sporting activities, Fitzpatrick phototype, ethnic origin, etc. It would be interesting to know if there is a clinical history of the degree of sun exposure, clinical signs and/or if there are histologic indirect markers of the degree of sun exposure, such as the degree of solar elastosis.

Thank you very much for this insightful comment. We agree that multiple factors influence sun exposure levels, which in turn can lead to different degrees of sun damage at the same anatomical site across different donors. However, collecting a complete catalog of relevant factors is challenging. While we have some information on relevant factors, it is incomplete. Nonetheless, we have updated Table S2 with as much clinical information as we have on each donor, but person-to-person correlative analyses will need to be the focus of a future manuscript performed properly as a case/control analysis.

The reviewer recommended a clever way to cut through these challenges by measuring solar elastosis. While this was feasible for the squamous cell carcinomas (discussed below), it was not feasible for the normal skin samples. Normal skin biopsies were typically 3mm punches, and we used the entirety of each tissue for single-cell studies.

Therefore, in the current version of the manuscript, we acknowledge, as a limitation of this study, that precise measurements of cumulative sun exposure are not available. (line 421-429)

- Ethnicity could be deduced from genomic data. Some of the data should be interpreted based on this new information on some objective measure of sun exposure. If not, this aspect should be considered a work limitation, and the authors should explain how much it could affect the results.

In response to the reviewer's comment, we have inferred ancestry from genomic data of all donors in Table S2. The resulting PCA plot has been added to Figure S2, and for the reviewer's convenience is shown here:

Based on this analysis, 14 out of 15 donors were confirmed to be of European ancestry. One donor (D21) was admixed American ancestry. Of note, donor 21 self-identified as Hispanic, so it was not a surprise that we had one non-Caucasian donor in our cohort. The sample size of cells from non-Caucasian donors is too small to draw definitive conclusions, and therefore, as suggested by the reviewer, we acknowledge this as a limitation to be addressed in future studies. (line 102-105, 509-522).

-On page 2, the last paragraph, it says, "individual keratinocytes from physiologically normal human skin." I think it could say, "individual keratinocytes from clinically normal human skin." One could also say histologically normal if histology had been analyzed.

Thank you very much for this comment. We have incorporated the more appropriate term as the reviewer has advised. (line 67)

-On page 4, reference 29 is from a review article and not the one with research data

We appreciate the reviewer's feedback and have added additional citations referring to studies published by Bowen et al. (PMID: 12535197) and Young et al. (PMID: 12856704, 3317295), which demonstrate that keratinocytes are more susceptible to UV-induced apoptosis, while melanocytes are relatively more resistant. Of note, the studies by Young et al. are classified as reviews, but they show primary data of apoptotic keratinocytes after sunburn. These studies by Young et. al. appear to be the earliest and most prominent citations to sunburn cells, but please let us know if we are missing any key literature.

-In methods, the available SCC clinicopathological features should be described (i.e., biopsy type, clinical size, thickness, degree of differentiation, etc.). It seems that most of the samples were taken by tissue saucerization and are well-differentiated cSCC. With this new information, evaluate whether modifying or clarifying any of the statements is necessary. For example, the one related to the cSCC differentiation spectrum in the spatial transcriptomic data analysis.

Thank you for this suggestion. We have added this information to the methods and Table S4. (line 696-700)

We revisited and confirmed the clinicopathological features of SCC, including those mentioned above by the reviewer, with the assistance of dermatopathologists. The excisional biopsy samples exhibited solar elastosis graded between 2+ and 3+. Biopsy sizes varied based on the clinical requirements of the procedure, and Breslow thickness ranged from 1.1 to 2.2 mm. As predicted by the reviewer, all squamous cell carcinomas were well differentiated, which we will acknowledge as a limitation in the manuscript. (line 307-314)

-On page 5, the authors suggest that in 4 cases, there were no mutations exclusive to the squamous cell, implying a single population of clonally related cells spanning both dissected tissue areas, with no identifiable mutations accounting for the progression to invasive carcinoma. For these cases, the actinic keratosis

histology may represent an extension of the squamous cell carcinoma rather than a distinct precursor lesion. I would like you to clarify this idea better. Are there other possibilities? (ie. Epigenetics, methodologic bias, non-detected mutations on cSCC, etc).

The reviewer raised an insightful point regarding the heterogeneity in morphology within keratinocyte neoplasms. There is a possibility that the histologically distinct areas are genetically different, despite the fact that we observe no differences. We would need to perform whole genome sequencing to say, with complete certainty, that there are no genetic differences. However, our panel covers 5 megabases of DNA, including all of the well-established drivers of SCC, so if there are genetic differences, they would be modest (likely less than 0.2 mutations/Mb, which is small in comparison to the overall mutation burden of 25 mutations/Mb observed in a typical cSCC).

Instead, we believe that the morphological heterogeneity within these keratinocyte cancers is driven by epigenetic mechanisms and the tumor microenvironment (TME). The differences in cell morphology, as identified by dermatopathologists, likely reflect distinct differentiation states. We have made edits to reflect this possibility in the main manuscript. (line 219-220)

-On page 5, the authors show that higher phospho-MAPK signalling was observed in the squamous cell carcinoma (Fig. 3H), consistent with the CBL mutation. I think it is a very indirect way to show the effect of CBL mutation. MAPK ERK could be activated for many other different reasons. Please comment on this point and defend your position

The reviewer raises a fair point. The CBL mutation is a known RASopathy mutation, and therefore would be predicted to activate MAPK/ERK signaling. However, other mutations and/or microenvironmental factors may also be contributing. Therefore, we have tempered this conclusion in the revised manuscript. (line 230-232, 243-244)

- Regarding figure 6. Why does Hippo appear in red? (FAT1 is considered a tumour suppressor gene, and Hippo ON controls YAP/TAZ, inducing sequestering in the cytoplasm and degradation by the proteasome)

Thank you very much for this insightful comment. We initially kept the Hippo signaling red to imply final activation of YAP/TAZ transcriptional coactivators (by lack of phosphorylation and consequent nuclear translocation for transcriptional activity). However, as the reviewer rightly pointed out, Hippo signaling must be "OFF" for YAP/TAZ activation. This clarification has also been incorporated into the manuscript. We have made the changes both in the introduction (line 55) and discussion (line 344).

Reviewer #3 (Remarks to the Author):

This manuscript presents a comprehensive multi-omic analysis of the genetic evolution of cutaneous squamous cell carcinoma (cSCC). The authors utilize single-cell and spatial transcriptomic techniques to track the progression from normal keratinocytes to actinic keratosis and finally to cSCC. This approach provides novel insights into the clonal architecture of cSCC development and identifies key mutational events and gene expression changes during this process. The findings challenge the traditional view of cSCC development, revealing unexpected complexities in clonal relationships, the role of specific mutations (TP53 and NOTCH1), and the spatial heterogeneity of immune responses within tumors. The identification of frequent "collisions" of unrelated neoplasms is particularly noteworthy and is in accordance with the concept of field cancerization. The manuscript is well-written and the results are presented clearly, with appropriate use of figures and supplementary materials.

Specific comments:

While the number of samples analyzed is substantial, a larger sample size, particularly for the spatial

transcriptomics, would strengthen the conclusions. The relatively low number of cases with a clear precursor-lesion relationship limits the power of those specific analyses. This should be commented.

Thank you for the feedback. As the reviewer notes, we do sequence a sizable number of individual cells, but the number of tumors with spatial transcriptomics data is limited. Spatial transcriptomics is still a relatively new technology, and there are very few spatial transcriptomic datasets (possibly none?) covering cancer/pre-cancer pairs with known phylogenies. The limiting step for these analyses was finding these informative cases – when we started this study, we did not expect that collisions of unrelated actinic keratoses and squamous cell carcinomas would predominate. Given the rarity of these tumors, we do believe these findings are worth including here. It would take considerable effort, going beyond the scope of this particular study, to substantially increase the sample size, but this is an ongoing priority for our laboratory. As such, we have deemphasized these finding by limiting the spatial transcriptomic data to a single main figure and reducing the text allocated to these results.

Identification of loss-of-function mutations of TP53, NOTCH1, NOTCH2, and CDKN2A and gain-of-function mutation of the TERT promoter in cSCC are in accordance with previous observations and not novel. However loss-of-function mutations in ARID2 and CBL in cSCC are interesting and should be discussed further e.g. in terms of their functional role in cSCC progression. Also the role of mutations resulting in the activation of MAPK pathway, e.g. in HRAS should be discussed.

These are great suggestions. The CBL mutation affects a mutational hotspot seen in patients with a RASopathy syndrome, known as Noonan syndrome. The ARID2 mutation is one of many ways to disrupt the SWI/SNF chromatin remodeling complex. We now describe the pathways effected by the ARID2 and CBL mutations, and since these pathways are recurrently altered at the transition from actinic keratosis to squamous cell carcinoma, we also discuss, more broadly, how loss of SWI/SNF and gain of MAPK signaling might drive this transition. (line 230-232, 243-244, 344-356)

The authors acknowledge the limitations of clonal expansion in introducing bias and the challenges of accurately calling somatic mutations in individual cells. Discussion of potential biases inherent in the spatial transcriptomics data is also warranted.

Thank you for the suggestion. We have added a sentence describing the main limitations of spatial transcriptomics in comparison to bulk-cell RNA-sequencing (line 284-286). In particular, the coverage of spatial transcriptomics data is typically lower, and it is difficult (or even impossible, depending on the platform) to resolve full-length transcriptional sequences, which limits information on splicing. Nonetheless, we felt the spatial information outweighed these limitations for this particular study.

The authors should further discuss the clinical implications of the results for diagnosis, prognosis, and treatment strategies.

We appreciate the feedback and have updated the discussion section to discuss these implications. (line 382-387)

Reviewer #4 (Remarks to the Author):

The manuscript has generated genomic profiles of multiple skin cell types, including keratinocytes, melanocytes, and fibroblasts across normal skin, precancerous lesions, and squamous cell carcinoma samples. The authors have shown that keratinocytes can cope with UV well, with fewer mutations accumulating than melanocytes. They have also shown the genomic path for the evolution of squamous cell carcinoma, especially by comparing the precancerous lesion and cancer area taken from the same patients.

Finally, they match their finding with spatial transcriptomics analysis on the same tissue, which shows the enrichment of activated/exhausted immune cells in SCC area.

One of the most interesting and novel findings from this study is the low mutational burden of keratinocytes. While the authors have suggested a potential mechanism for this, there's no experimental validation, which limits the impact of the current finding. As the authors have set up the culture conditions for these cell types, the hypothesis can be tested by UV exposure and clonal evolution analysis *in vitro*.

Thank you for the suggestion. In this revision, we performed *in vitro* irradiation of melanocytes and keratinocytes with the exact same doses of simulated solar UV radiation (line 135-148, 666-691). These studies were performed on neonatal cells, derived from human foreskin, as these cells would have uniform and low mutation burdens at baseline, allowing us to detect changes in mutation burden following UV irradiation. The results are shown in figure S4 and reprinted here for the reviewer's convenience.

We confirmed what others have previously shown (PMIDs: 12535197, 12856704, 3317295 and 10219070) – keratinocytes are more sensitive to UV radiation than melanocytes. However, we went further by mutationally profiling individual cells after irradiation. To our surprise, the surviving keratinocytes had more mutations than the surviving melanocytes. These findings are fascinating but raise new questions.

How is it possible for keratinocytes in the skin to have, on average, fewer mutations than melanocytes when keratinocytes accumulate mutations *in vitro* at a faster rate than melanocytes? A major limitation to 2-dimensional irradiation is that it does not model mutation accumulation in a natural microenvironment or on an appropriate time scale (ideally decades of cumulative exposure rather than a single acute exposure). For example, we have a separate study where we find populations of melanocytes with low mutation burdens in heavily sun damaged skin, and we show that they likely spent most of their lives in hair follicles before migrating to the epidermis (<https://www.biorxiv.org/content/10.1101/2025.02.07.637114v1>). We would not be surprised if a similar mechanism is occurring in the setting of keratinocytes. As another possibility, keratinocytes in the skin maintain supranuclear "caps" of melanin, which protects their genomes *in situ*, but this protection would not be present in 2-dimensional monocultures of keratinocytes.

Nevertheless, the 2-dimensional irradiation results are informative because they rule out some possibilities. For example, the 2-dimensional irradiation work suggests that keratinocytes are not superior at DNA repair than melanocytes.

[FIGURE REDACTED]

Supranuclear cap of melanin.
 Keratinocytes receive melanin from melanocytes and deposit the pigment above their nuclei to protect against UV radiation. Supranuclear caps of melanin are not present in monocultures of keratinocytes.

In light of these results, we have altered some of our speculation about how keratinocytes maintain low mutation burdens in human skin. It would be interesting to dive even deeper into the mechanisms, but such experiments go well beyond the scope of this study. Indeed, the data in figure S4, alone, was not trivial to produce. Each data point encompasses an individual cell that was clonally expanded and had whole exome/transcriptome sequencing performed. Extending this work to appropriate *in vivo* models would take several years of effort.

In addition, the authors have tried to combine the spatial transcriptomics (10X Visium) with the clonal evolution. However, the current scope of spatial analysis is of low resolution, with simple sketches with gene set signature scores. Except for the fact that there are more activated/exhausted immune cells enriched around the SCC area compared to AK area, which is already well known and expected, the spatial transcriptomics do not offer any advancement in our understanding of tumor microenvironment evolution.

This comment was also raised by the other reviewers, and we agree that the sample size of spatial transcriptomics is low, but this is because these are rare tumors. It was more difficult than we anticipated to find examples of squamous cell carcinomas that were adjacent to an actinic keratosis and genetically confirmed to have evolved from the actinic keratosis. There are not many (possibly none?) spatial transcriptomic datasets of cancer/pre-cancer pairs with well-defined mutational phylogenies in any cancer subtype. Given the rarity of these datasets, we maintain our belief that this data is worth reporting, but in the revision, we have reduced the emphasis on these results by shortening the text in this section.

Nonetheless, the reviewer is correct that the spatial transcriptomics data is comparatively less interesting than the other findings in our manuscript. As such, we limit these findings to one main figure and have reduced the text dedicated to these results.